

# From data compilation to model validation: a comprehensive analysis of a full deep-sea ecosystem model of the Chatham Rise

Vidette L. McGregor[1,2], Peter L. Horn[1], Elizabeth A. Fulton[3] and Matthew R. Dunn[1]

[1] Fisheries, National Institute of Water and Atmospheric Research Limited, Wellington, New Zealand
[2] School of Biological Sciences, Victoria University of Wellington, Wellington, New Zealand
[3] Marine Ecosystem Modelling and Risk Assessment, CSIRO Marine Research, Hobart, TAS, Australia

## ABSTRACT

The Chatham Rise is a highly productive deep-sea ecosystem that supports numerous substantial commercial fisheries, and is a likely candidate for an ecosystem based approach to fisheries management in New Zealand. We present the first end-to-end ecosystem model of the Chatham Rise, which is also to the best of our knowledge, the first end-to-end ecosystem model of any deep-sea ecosystem. We describe the process of data compilation through to model validation and analyse the importance of knowledge gaps with respect to model dynamics and results. The model produces very similar results to fisheries stock assessment models for key fisheries species, and the population dynamics and system interactions are realistic. Confidence intervals based on bootstrapping oceanographic variables are produced. The model components that have knowledge gaps and are most likely to influence model results were oceanographic variables, and the aggregate species groups 'seabird' and 'cetacean other'. We recommend applications of the model, such as forecasting biomasses under various fishing regimes, include alternatives that vary these components.

# INTRODUCTION

The goal of incorporating a holistic approach to understanding the system-wide repercussions of how we manage our marine resources is admirable and ambitious (*Long, Charles & Stephenson, 2015*; *Link & Browman, 2017*). Ecosystem based management (EBM) requires a range of tools, often including ecosystem models (*Smith et al., 2017*; *Stecken & Failler, 2016*). Within ecosystems there are many processes at play, and the models developed to support EBM vary in scope and complexity (*Plagányi, 2007*; *Fulton, 2010*; *Collie et al., 2016*). End-to-end ecosystem models that can deal with bottom-up and top-down system controls have become popular for exploring

Corresponding author
Vidette L. McGregor,
vidette.mcgregor@niwa.co.nz

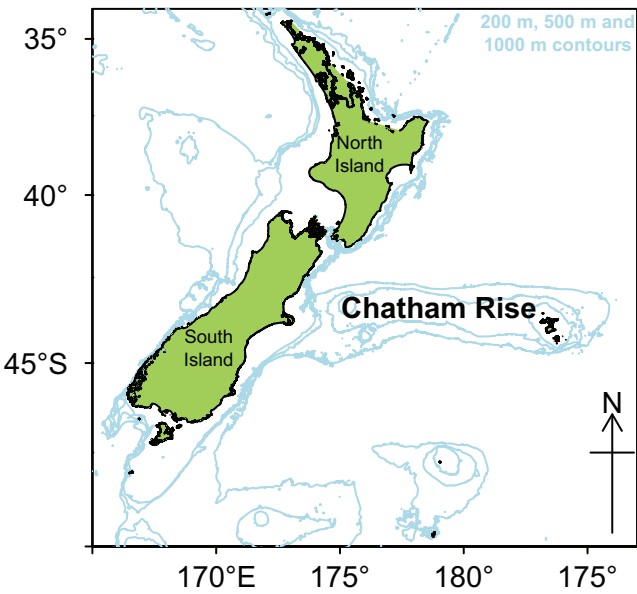

**Figure 1** Map of New Zealand with Chatham Rise marked, including 200, 500, and 1,000 m isobaths.

scenarios involving human induced impacts including fishing and climate change
(*Rose, 2012*).

The body responsible for fisheries management in New Zealand, Fisheries New Zealand,
is seeking to move away from single species management towards a more ecosystem
approach, both to fulfil Fisheries Act obligations and Marine Stewardship Council (MSC)
expectations (*Ministry for Primary Industries, 2008*; *Marine Stewardship Council, 2014*).
The Chatham Rise is the location of several nationally important MSC certified
fisheries (*Deepwater Group, 2018*), and a growing understanding of trophic interactions
exists there (*Stevens, Hurst & Bagley, 2011*; *Dunn et al., 2009*).

Chatham Rise is a submarine ridge running eastwards for about 1,000 km from the east
coast of South Island, New Zealand, rising up from depths of about 3,000 m, to about
50 m at the western end, and sea level around the Chatham Islands at the eastern
end (Fig. 1). The subtropical front (STF), a relatively broad permanent feature where
warmer, more saline, and nutrient poor subtropical water from the north meets nutrient
rich subantarctic water from the south, extends up the east coast of South Island,
and then eastwards along Chatham Rise (*Heath, 1985*; *Uddstrom & Oien, 1999*).
The demersal fish assemblage on Chatham Rise has the highest fish species richness in
New Zealand waters (*Leathwick et al., 2006*). The range of habitats and depths, and the
influence of the STF, are expected to provide a wide variety of foraging opportunities
for demersal and pelagic organisms.

The Chatham Rise is perhaps New Zealand's most productive fishing ground.
It supports substantial commercial fisheries for finfish and invertebrates, with notable
examples being: trawl fisheries for hoki (*Macruronus novaezelandiae*), orange roughy
(*Hoplostethus atlanticus*), hake (*Merluccius australis*), and black and smooth oreos
(*Allocyttus niger*, *Pseudocyttus maculatus*); a longline fishery for ling (*Genypterus*

*blacodes*); and a potting fishery for rock lobster (*Jasus edwardsii*) (*New Zealand Ministry for Primary Industries, 2014*).

Analyses of trawl survey series and commercial fishery catch rates have shown that marked variations over time have occurred in the relative abundance of some common species on Chatham Rise, for example, hoki, hake, orange roughy, scampi (*Metanephrops challengeri*), and rock lobster (*Maunder & Starr, 1995*; *Dunn, Anderson & Doonan, 2008*; *Stevens et al., 2017*). Some factors driving these fluctuations have been identified (i.e. high exploitation levels, variation in recruitment), but there will certainly be other physical and biological factors that will influence animal behaviour and survivability, resulting in changes to the ecosystem. A knowledge of how particular biological and ecological changes could affect the abundance and distribution of species will usefully inform the management of those species.

In an ecosystem, nothing exists independently. When assessing biological risks, it is difficult to conceptualise risk to the whole system. A system-level model within which different scenarios can be explored is an extremely valuable tool for gaining conceptual understanding of economic and biological risks for a whole system, as well as for individual parts.

Atlantis is an end-to-end ecosystem modelling approach that can be used to create an environment in which different scenarios can be played out to test for different results and learn how a system may be reacting to changes within it. Reviewed as one of the best modelling frameworks for exploring 'what-if' type questions (*Plagányi, 2007*), it includes the ability to compare social, conservation, and economic outcomes. With sufficient data, this modelling approach can be extremely useful for management strategy evaluation (*Plagányi, 2007*), and has been applied to multiple marine systems (from single bays to millions of square kilometres) in Australia, the US, Europe, and South Africa (*Savina et al., 2005*; *Fulton, Smith & Smith, 2007*; *Link, Fulton & Gamble, 2010*; *Ainsworth, Schirripa & Morzaria-Luna, 2015*; *Smith, Fulton & Day, 2015*; *Sturludottir et al., 2018*; *Ortega-Cisneros, Cochrane & Fulton, 2017*). Atlantis is a deterministic simulation model such that for a given parameter set and model specification, the model outputs are identical. Atlantis models are too complex to statistically fit to observations, although subsets of key parameters can be estimated using statistical methods outside of the model. Analysing and understanding the model dynamics and potential weaknesses is essential before the model can be used to learn about the system.

In this paper, we describe the first end-to-end ecosystem model for the Chatham Rise, New Zealand. We present analyses of the model, comparing its state and dynamics to current knowledge. We identify and assess the likely influence of current knowledge gaps and uncertainties.

In developing such models, knowledge gaps become evident, and we are provided with the opportunity to analyse the importance of these gaps, thus guiding direction of future research. The model was assessed for single species dynamics and inter-species connectivity. We conducted a skill assessment on species groups for which we have surveys capable of indexing abundance, and compared biomass trends as the model responded

to historical fishing for species groups that have stock assessments or reliable catch per unit effort (CPUE) indices. We simulated changes in biomass for each species group and analysed responses throughout the system. This latter part formed the basis for analysing influence and importance of knowledge gaps, and where a species group performed poorly in the skill assessment it often highlighted a knowledge gap.

## METHODOLOGICAL APPROACH

The process of developing this model was not linear, but rather iterative and incremental. There were five main stages to the development, each of which was re-visited until we were satisfied with the performance of the model and our understanding of its dynamics. The main stages can be summarised as:

(1) Data and model inputs were collated and defined.
(2) The base historical model was calibrated without fishing such that this model had stable biomass trajectories over the 1900–2016 model period, realistic diets, growth rates, natural mortalities.
(3) Sensitivity analyses were carried out with respect to oceanographic variables and simulations aimed at understanding connectivity and influence between the species functional groups.
(4) Fishing was included in the model using forced catch removals.
(5) Skill assessment and comparisons to abundance indices and biomass estimates were carried out.

'Model design, Calibration, Sensitivity analyses, Fishing, Skill assessment cover' each of these five main stages, followed by 'Bringing it together': Bringing it together, which discusses some of the implications of the models' performance, dynamics, and data gaps.

## MODEL DESIGN

An Atlantis model simulates the ecosystem through time, calculating each new state based on the previous state and the events of the current timestep. This section describes the physical, biological, ecological, and fishing components of the Chatham Rise Atlantis Model (CRAM). Further details on Atlantis can be found in the Atlantis user manual (*Audzijonyte et al., 2017*).

### Model area

The Chatham Rise Atlantis model area comprises waters from the shore-line around Chatham Islands (but excluding estuaries on the islands) to depths of 1,300 m along the Chatham Rise (Fig. 2). The western boundary of the area is defined as the 400 m contour on the western edge of the Mernoo Gap, a trough that separates the Chatham Rise from the coastal shelf off the mid-east coast of South Island.

An Atlantis model requires the modelled region to be split into polygons and depth layers. Each polygon/depth layer is referred to as a cell. The intention of the splits is to capture important aspects of the region but at a simplified level such that modelling the region over many years becomes possible. If we were modelling a smaller temporal
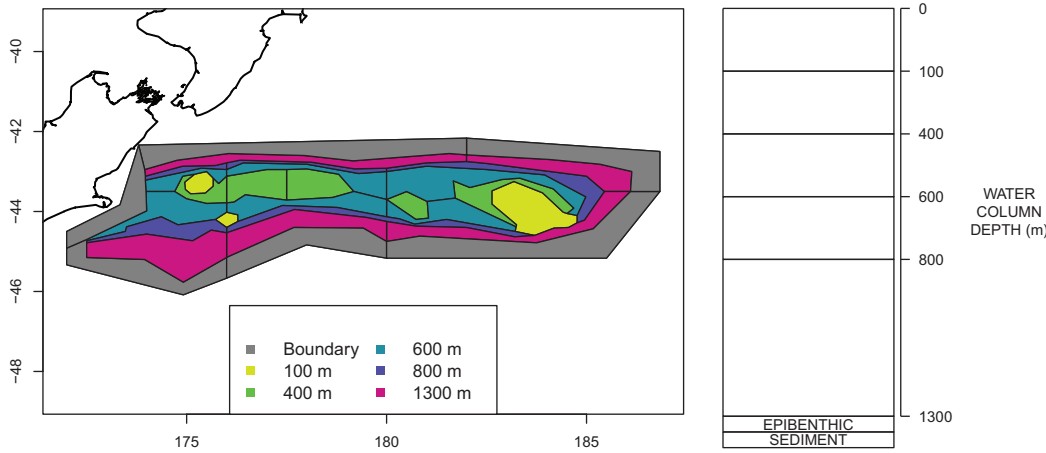

**Figure 2** Polygons as defined for CRAM with maximum depths for each polygon shown by colour (left) and depth layer bins (right).

scale, we may have considered a finer spatial scale. The polygons within the modelled area are referred to as dynamic polygons, and these are surrounded by non-dynamic polygons which define the boundary conditions for the modelled domain.

Several investigations of fish communities or fish species richness indicated that the division of the Chatham Rise into polygons for Atlantis modelling should occur primarily based on depth categories, with the northern and southern slopes separated (owing to the different water masses and fish communities to the north and south of the STF), and with some longitudinal differentiation as well. Species communities were found to group in adjacent depth-defined strata, but with differences between depths on the northern and southern Rise, as well as some longitudinal differentiation (*Tuck, Cole & Devine, 2009*).

A large amount of data on the abundance and distribution of demersal fish and invertebrate species has been collected from the series of trawl surveys of depths 200–800 m on Chatham Rise in January annually from 1992 to 2014 (*Livinston et al., 2002*; *Stevens et al., 2017*). Some of the more recent surveys in the series also included strata to depths of 1,300 m (*Stevens et al., 2017*). The survey area was stratified by depth, latitude, and longitude. It was logical, therefore, to base the Atlantis model polygon boundaries on the trawl survey strata boundaries. This is also helpful for informing the model spatially based on trawl surveys. Consequently, the model area was divided into 23 dynamic polygons based on bottom depth bins (<200 m, 200–400 m, 400–600 m, 600–800 m, 800–1,300 m), with bins deeper than 400 m separated into northern and southern Rise polygons, and with longitudinal separation (where trawl survey strata allowed) aimed at producing western, central, and eastern polygons. The dynamic polygon area is surrounded by six additional non-dynamic polygons which allows for the exchange of water, nutrients and biota into and out of the dynamic model domain. The final configuration of the dynamic and non-dynamic polygons is shown in Fig. 2.

All model polygons are further divided into water column depth layers, ranging from one layer in some near-shore polygons to five layers for the deepest polygons.

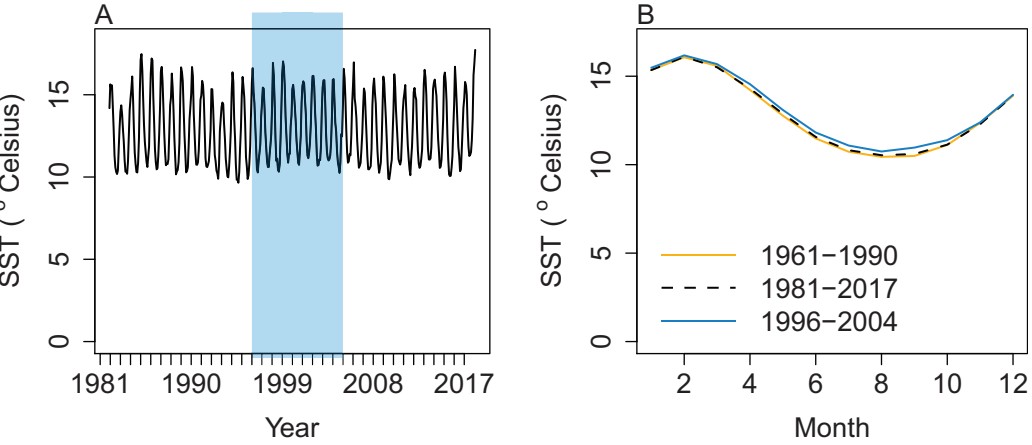

**Figure 3** Sea surface temperature (SST) (°Celsius) weekly averages for 1981–2017 with ROMS years 1996–2004 shaded blue (A) and mean SST by month (B) from the same data for 1981–2017 (black dashed line), with the subset from 1996 to 2004 (blue solid line), and additional historical SST data from 1961 to 1990, which were only available as monthly averages (orange solid line).

Depth layers are also defined in Fig. 2. Each box also contains one epibenthic and one sediment layer.

## Time

The model was run with a 35 year burn-in period (1865–1900) followed by a 115 year modelled period (1900–2015). The burn-in period allows for the model to adjust from potentially unstable initial conditions due to uncertainty of some of the parameters and age distributions for the age resolved groups, to a state, that is, more stable. A 35-year period was chosen as it covered initial fluctuations of most functional groups in the model. All results presented here are from the modelled period 1900–2015. The model used 12 hour timesteps to allow for changes in temperature, light and feeding patterns between night and day.

## Oceanography

Salinity, temperature, and water exchange between cells were forced in the Atlantis model using outputs from a Regional Oceanographic Modelling System (ROMS) model (*Hadfield, Rickard & Uddstrom, 2007*) that covered years 1996–2004. Water currents across each box face cause the horizontal movement of nutrients (such as ammonia and nitrate) available to primary producers. The speed and direction of currents influence the spatial distribution of plankton groups. Water temperatures influence biological processes such as respiration (*Hoegh-Guldberg & Bruno, 2010*). Based on sea surface temperatures, the ROMS years (1996–2004) look to be fairly representative of those properties from 1961 to 2017 (Fig. 3). The base model presented here repeated the available ROMS variables as a 9-year cycle. Averaging the ROMS variables was not sensible due to the water exchange between cells, as these change every 12-h timestep in strength and direction, and averaging them could easily result in implausible physical dynamics.

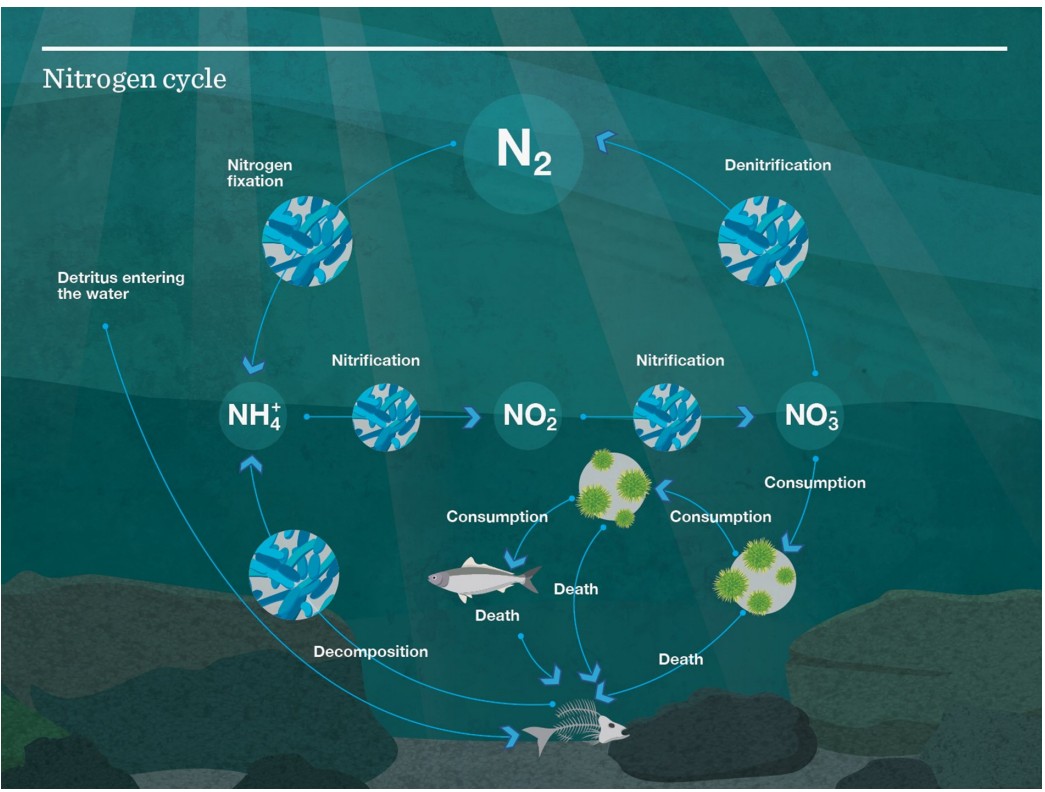

**Figure 4 Nutrient cycle as modelled in Atlantis.**

We ran sensitivities varying the order of ROMS years or repeating one ROMS year to help understand the effects of inter-annual oceanographic variability on this model.

## Nutrients

Atlantis models use nitrogen, an important and often limiting nutrient in marine systems (*Moore et al., 2013*), to track the transfer of energy throughout the system. The nitrogen cycle can be seen in Fig. 4. When biomass pools are tracked in the model, they are done so in mg N/m$^3$. When a fish (e.g.) eats another fish, it is nitrogen, that is, transferred up the food chain, with some nitrogen going to detritus and carrion, thus providing nitrogen to micro-organisms and filter feeders to fuel the cycle over again.

### Nutrient data

Oxygen ($O_2$), nitrates ($NO_3$), ammonium ($NH_4^+$), and silica ($SiO_2$) were simulated in the model, and required spatially defined initial conditions (values for each cell in the model domain). Table 1 has a summary of the data sources for these nutrients. We used values from the World Ocean Atlas (WOA) for initial conditions for nitrate values down to 500 m, oxygen down to the full model depth of 1,300 m, and silica down to 1,300 m. The WOA contains objectively analysed climatological fields of in situ oxygen, temperature, salinity, and some nutrients (*Locarnini et al., 2013*; *Zweng et al., 2013*; *Garcia et al., 2013a, 2013b*). $NO_3$ μ mol/m$^3$ were converted to mg N/m$^3$ by multiplying by 14 as the molecular mass of nitrogen is 14 g/mol.

**Table 1 Sources of data for oxygen, nitrates, ammonium, and silica.**

| Variable | Source | Depth (m) | Latitude | Longitude |
|----------|--------|-----------|----------|-----------|
| Oxygen | WOA | 1,300 | 42–47 S | 172 E–170 W |
| Oxygen | WOCE | 1,300 | 42.5 S | 180 E |
| Nitrate | WOA | 500 | 42–47 S | 172 E–170 W |
| Nitrate | WOCE | 1,300 | 42.5 S | 180 E |
| Silica | WOCE | 1,300 | 42.5 S | 180 E |
| Ammonium | NIWA survey | 0–50 | 43–46 S | 172–180 E |

Note:
WOA, World Ocean Atlas; WOCE, World Ocean Circulation Experiment; NIWA, National Institute of Water and Atmospheric Research.

World Ocean Circulation Experiment (WOCE) (*Deutsches Ozeanographisches Datenzentrum, Bundesamt Fur Seeschifffahrt Und Hydrographie G, 2006*) data were used for nitrates at depths greater than 500 m, which were not covered by WOA. WOCE data were also used to compare values for oxygen, to inform initial conditions for silica, and to compare with salinity, temperature and chlorophyll *a*.

Ammonium values were available from National Institute of Water and Atmospheric Research (NIWA) oceanographic surveys, but only down to 50 m. This was not too concerning as ammonium is a small component of the nitrogen budget.

## Species groups

CRAM uses 53 functional groups to model the biological processes. Of these 53 groups, 15 vertebrates, and one invertebrate comprised single species; all other groups comprised two or more species. The main component species of the groups are shown in Tables 2–5. All vertebrate groups and five invertebrate groups were modelled with age-structure using up to 10 age-classes and varying number of years per age-class, depending on the longevity of the primary species in the group. Within each age-class, the model simulated numbers of individuals and the average weight (mg N) of individuals within each age class. Weights were split into structural ($S_N$) and reserve ($R_N$) components following the definition in *Broekhuizen et al. (1994)* where reserve weight is the part that can be used during periods of starvation, which includes flesh, fat, reproductive components, and other soft tissue. Primary producers and remaining invertebrate groups were modelled as biomass pools (mg N/m$^3$) with no age-structure.

### Initial conditions and biological parameters for species groups

Initial biomasses for each species group were estimated using a single species stochastic stock assessment model, CASAL (*Bull et al., 2012*). Biomass estimates for the entire Chatham Rise were derived by using known biological parameters and a catch history to project back from an absolute abundance estimate in 2003. Values of relative abundance were available for most species groups from trawl surveys conducted annually from 1992 to 2014 (see *O'Driscoll et al., 2011*). For each survey, these abundance estimates were converted to absolute values using trawl catchability quotients (specific to each group) derived by our expert opinion, as fisheries scientists with experience dating back more than 30 years. Estimated absolute abundance for each group in 2003 (the midpoint of

**Table 2 List of functional vertebrate groups for CRAM.**

| Name | Main species | Lifespan (years) |
| --- | --- | --- |
| Baleen whales | Southern right whales (*Eubalaena australis*) | 80 |
| Basketwork eel | Basketwork eels (*Diastobranchus capensis*) | 30 |
| Baxters dogfish | Baxter's dogfish (*Etmopterus baxteri*) | 50 |
| Ben fish deep | Four-rayed rattail (*Coryphaenoides subserrulatus*) | 20 |
| Ben fish shal | Oblique banded rattail (*Coelorinchus aspercephalus*) | 10 |
| Black oreo | Black oreo (*Allocyttus niger*) | 120 |
| Bollons rattail | Bollons' rattail (*Caelorinchus bollonsi*) | 20 |
| Cetacean other | Primarily sperm & pilot whales & dolphins | 30 |
| Dem fish pisc | Giant stargazer (*Kathetostoma giganteum*) | 20 |
| Elasmobranch invert | Primarily skates & dogfish | 20 |
| Elasmobranch pisc | Primarily semi-pelagic sharks | 50 |
| Epiben fish deep | Spiky oreo (*Neocyttus rhomboidalis*) | 100 |
| Epiben fish shal | Common roughy (*Hoplostethus atlanticus*) | 10 |
| Ghost shark | Dark ghost shark (*Hydrolagus novaezealandiae*) | 20 |
| Hake | Hake (*Merlucciidae*) | 30 |
| Hoki | Hoki (*Macruronus novaezelandiae* | 20 |
| Javelinfish | Javelinfish (*Coelorinchus australis*) | 10 |
| Ling | Ling (*Molva molva*) | 30 |
| Lookdown dory | Lookdown dory (*Cyttus traversi*) | 30 |
| Mackerels | Slender jack mackerel (*Trachurus murphyi*) | 30 |
| Orange roughy | Orange roughy (*Hoplostethus atlanticus*) | 120 |
| Pelagic fish lge | Southern bluefin tuna (*Thunnus thynnus*) | 20 |
| Pelagic fish med | Barracouta (*Thyrsites atun*) | 10 |
| Pelagic fish sml | Myctophids (*Myctophidae*) | 4 |
| Pinniped | NZ fur seal (*Arctocephalus forsteri*) | 20 |
| Reef fish | Blue cod (*Parapercis colias*) | 20 |
| Seabird | Seabirds & shorebirds | 20 |
| Seaperch | Seaperch (*Helicolenus* spp.) | 50 |
| Shovelnosed dogfish | Shovelnosed dogfish (*Deania calcea*) | 40 |
| Smooth oreo | Smooth oreo (*Pseudocyttus maculatus*) | 100 |
| Spiny dogfish | Spiny dogfish (*Squalus acanthias*) | 30 |
| Warehou | Silver, white & blue warehou | 20 |

Note:
Name is the species group name which is the same as the main species name for single-species groups but without punctuation. Lifespan is the assumed maximum number of years an individual in that group may live. Ben, benthic; Dem, demersal; invert, invertivore; pisc, piscivore.

the survey series) was taken as the mean from all the survey estimates. For each species group, the initial biomass estimate was distributed across polygons in proportion to the survey series estimates (i.e. the mean proportion of total biomass by polygon over the survey series). The distribution of biomass by depth layer in each polygon was derived using our expert opinion. Where there was no available catch history (e.g. seabirds), or no useful estimates of relative abundance from the trawl surveys (e.g. rock lobster),

**Table 3 List of functional invertebrate groups for CRAM.**

| Name | Description | Lifespan (years) |
|---|---|---|
| Arrow squid | Arrow squid | 2 |
| Benthic carniv | Benthic carnivores | |
| Carniv zoo | Planktonic animals (size 2–20 cm) | |
| Cephalopod other | Squid and octopus | 2 |
| Deposit feeder | Detritivores and benthic grazers | |
| DinoFlag | Dinoflagellates | |
| Filter other | Non-commercial benthic filter feeders | |
| Gelat zoo | Salps, ctenophores, jellyfish | |
| Invert comm herb | Paua and kina | 10 |
| Invert comm scav | Primarily scampi and crabs | 14 |
| Meiobenth | Benthic organisms (size 0.1–1 mm) | |
| MesoZoo | Planktonic animals (size 0.2–20 mm) | |
| MicroZoo | Heterotrophic plankton (size 20–200 μm) | |
| Rock lobster | Rock lobster | 12 |

Note:
Name is the species group name which is the same as the species name for single-species groups. Description includes main species. Lifespan is the maximum number of years an individual in that group may live. Those groups with no value for lifespan are modelled as biomass pools and hence do not have a lifespan defined as this is only relevant when modelling numbers. Zoo, zooplankton; Invert comm, commercial invertebrates; herb, herbivore; scav, scavenger.

**Table 4 List of functional phytoplankton and algae groups for CRAM.**

| Name | Description |
|---|---|
| Diatoms | Diatoms (large phytoplankton) |
| Macroalgae | Macroalgae |
| Microphytobenthos | Unicellular benthic algae |
| Pico-phytoplankton | Small phytoplankton |

Note:
Name is the species group name which is the same as the main species name for single-species groups. Description includes main species.

**Table 5 List of functional bacteria and detritus groups for CRAM.**

| Name | Description |
|---|---|
| Carrion | Dead and decaying flesh |
| Labile detritus | Organic matter that decomposes at a fast rate |
| Pelagic bacteria | Pelagic bacteria |
| Refractory detritus | Organic matter that decomposes at a slow rate |
| Sediment bacteria | Sediment bacteria |

Note:
Name is the species group name which is the same as the main species name for single-species groups. Description includes main species.

initial biomasses (and their distribution by model polygon) were estimated using our expert opinion. For age-structured groups, initial biomass estimates were assigned to age-classes using estimates of instantaneous natural mortality ($M$). Initial average weights

**Table 6 Biological parameters assumed for age-structured species groups.**

| Species group | VB growth | | | Length-weight | | $M$ | $h$ | Reference |
|---|---|---|---|---|---|---|---|---|
| | Linf (cm) | K | $T_0$ | $a$ | $b$ | | | |
| Arrow squid | 35 | 2.4 | 0 | 2.90E-02 | 3 | 4.6 | 0.8 | *Ministry for Primary Industries (2016)* |
| Baleen whales | | | | | | 0.01 | 0.5 | |
| Basketwork eel | 47.3 | 0.283 | −1.294 | 2.35E-03 | 3.25 | 0.19 | 0.8 | Trawl db |
| Baxters dogfish | 64.4 | 0.06 | −2.97 | 5.95E-03 | 3.068 | 0.08 | 0.3 | *Irvine, Stevens & Laurenson (2006a)* |
| Ben fish deep | 36 | 0.3 | −1.1 | 7.28E-03 | 2.632 | 0.2 | 0.8 | *Stevens et al. (2010)*, Trawl db |
| Ben fish shal | 38 | 0.3 | −1.1 | 2.35E-03 | 3.25 | 0.2 | 0.8 | *Stevens et al. (2010)*, Trawl db |
| Black oreo | 37 | 0.1 | −2 | 7.80E-03 | 3.27 | 0.044 | 0.75 | *Ministry for Primary Industries (2016)* |
| Bollons rattail | 47.3 | 0.283 | −1.294 | 2.35E-03 | 3.25 | 0.19 | 0.8 | *Stevens et al. (2010)* |
| Cephalopod other | 45 | 2.4 | 0 | 2.90E-02 | 3 | 4.6 | 0.8 | |
| Cetacean other | | | | | | 0.033 | 0.5 | |
| Dem fish pisc | 69.8 | 0.17 | −0.53 | 1.50E-02 | 3.01 | 0.19 | 0.8 | *Sutton (1999)*, *Ministry for Primary Industries (2016)* |
| Elasmobranch invert | 150.5 | 0.095 | −1.06 | 2.68E-02 | 2.933 | 0.135 | 0.3 | *Ministry for Primary Industries (2016)* |
| Elasmobranch pisc | 84.7 | 0.1065 | −4.56 | 1.50E-03 | 3.334 | 0.09 | 0.3 | *Irvine, Stevens & Laurenson (2006b)* |
| Epiben fish deep | 35.3 | 0.07 | −0.5 | 2.83E-02 | 2.9322 | 0.05 | 0.75 | *Stewart & Smith (1994)*, Trawl db |
| Epiben fish shal | 24 | 0.18 | −0.3 | 2.65E-02 | 2.9126 | 0.2 | 0.8 | Trawl db |
| Ghost shark | 97 | 0.09 | −1.17 | 2.02E-03 | 3.274 | 0.35 | 0.3 | *Ministry for Primary Industries (2016)* |
| Hake | 95.9 | 0.279 | 0.05 | 2.00E-03 | 3.288 | 0.19 | 0.8 | *Horn (2013)* |
| Hoki | 100.8 | 0.164 | −2.16 | 4.79E-03 | 2.89 | 0.275 | 0.75 | *McKenzie (2016)*, *Ministry for Primary Industries (2016)* |
| Invert comm herb | 155 | 0.15 | 0 | 3.00E-05 | 3.303 | 0.15 | 0.8 | *Breen, Kim & Andrew (2003)* |
| Invert comm scav | 50 | 0.25 | 0 | 3.73E-04 | 3.145 | 0.2 | 0.8 | *Tuck (2016)* |
| Javelinfish | 51.2 | 0.216 | −1.618 | 1.38E-03 | 3.13 | 0.35 | 0.8 | *Stevens et al. (2010)* |
| Ling | 135.2 | 0.105 | −0.72 | 1.07E-03 | 3.336 | 0.14 | 0.84 | *McGregor (2015)* |
| Lookdown dory | 50 | 0.075 | −1 | 2.35E-02 | 2.97 | 0.15 | 0.8 | *Stewart & Smith (1994)*, *Ministry for Primary Industries (2016)* |
| Mackerels | 74.25 | 0.111 | −0.811 | 2.38E-02 | 2.7671 | 0.3 | 0.7 | *Cubillos et al. (1998)*, *Kochkin (1994)* |
| Orange roughy | 37.2 | 0.065 | −0.5 | 9.21E-02 | 2.71 | 0.045 | 0.75 | *Ministry for Primary Industries (2016)* |
| Pelagic fish lge | 182 | 0.205 | 0 | 1.88E-02 | 3.0078 | 0.2 | 0.8 | *Fournier et al. (1990)*, *Ministry for Primary Industries (2016)* |
| Pelagic fish med | 85.2 | 0.298 | −0.45 | 7.40E-03 | 2.94 | 0.3 | 0.7 | *Horn (2002)*, *Ministry for Primary Industries (2016)* |
| Pelagic fish sml | 7 | 0.8 | 0 | 1.30E-02 | 2.81 | 1.58 | 0.7 | *Young et al. (1988)*, Trawl db |
| Pinniped | | | | | | 0.07 | 0.5 | |
| Reef fish | 51.7 | 0.087 | −1.7 | 1.91E-02 | 2.9818 | 0.14 | 0.8 | *Ministry for Primary Industries (2016)* |
| Rock lobster | 85 | 0.15 | 0 | 4.16E-03 | 2.935 | 0.12 | 0.8 | *Ministry for Primary Industries (2017)* |
| Seabird | | | | | | 0.11 | 0.5 | |
| Seaperch | 45.6 | 0.08 | −0.8 | 7.77E-03 | 3.22 | 0.07 | 0.8 | *Paul & Horn (2009)*, *Ministry for Primary Industries (2016)* |
| Shovelnosed dogfish | 106.4 | 0.106 | −0.384 | 1.58E-03 | 3.192 | 0.13 | 0.3 | *Clarke, Connolly & Bracken (2002)*, Trawl db |
| Smooth oreo | 46 | 0.07 | −1.5 | 3.05E-02 | 2.885 | 0.063 | 0.75 | *Ministry for Primary Industries (2016)* |

(Continued)

| Species group | VB growth | | | Length-weight | | M | h | Reference |
|---|---|---|---|---|---|---|---|---|
| | Linf (cm) | K | $T_0$ | a | b | | | |
| Spiny dogfish | 104.8 | 0.093 | −3.17 | 1.30E-03 | 3.2639 | 0.2 | 0.3 | *Hanchet (1986)*, *Beentjes & Stevenson (2009)* |
| Warehou | 53.1 | 0.37 | −0.88 | 8.28E-03 | 3.214 | 0.25 | 0.8 | *Horn & Sutton (1996)*, *Ministry for Primary Industries (2016)* |

**Note:**

VB, von Bertalanffy; M, instantaneous natural mortality rate; h, steepness value for the Beverton–Holt stock recruitment relationship. Length-weight parameters are: $W = aL^b$ (weight $W$ in g, length $L$ in cm). Where Reference is 'Trawl db' some data have been derived from the NIWA trawl survey database (see *Mackay, 2000*). Species group matches 'Name' in Tables 2 and 3 and are without punctuation.

at age were calculated using von Bertalanffy growth and length-weight conversion parameters. Values used for these parameters are in Table 6. Weights at age were split into reserve and structural components using ratio $R_N$:$S_N$ = 2.5:1. This allows for an individual's body mass to decrease by approximately 70% before starving, which is within the 60–80% range suggested by *Broekhuizen et al. (1994)*.

All age-structured groups were modelled with Beverton-Holt recruitment, the steepness ($h$) values for which are in Table 6. These values are not ever well known, and scenarios explored using this model should consider sensitivities for these.

## Predation

Simulated predation was a four-step process that occurred within each cell and at each timestep. From the predator's perspective the steps modelled can be summarised as: (1) Am I allowed to eat it?, (2) Is it in the same place at the same time as me?, (3) Does it fit in my mouth?, (4) How much can I eat? Full details are in the Atlantis User's Guide (*Audzijonyte et al., 2017*). Step 4 uses a feeding functional response, of which there are 12 options currently available in Atlantis. We have applied the Holling Type II functional response to all age-structured species groups in this model, thus influencing the amount of prey consumed by prey abundance, and the predators search rate and handling time.

Diets of each species group were summarised in categories Algae, Bacteria, Bird, Cetacea, Coelenterate, Crustacean, Detritus, Echinoderm, Elasmobranch, Microzooplankton, Mollusc, Phytoplankton, Polychaete, Teleost, and Tunicate similar to that done in the diet study of *Stevens, Hurst & Bagley (2011)* (Fig. 5). While this summary misses the temporal, spatial, age, and size components of the predator–prey interactions, it is useful to check overall diets. For example, warehou and smooth oreos eat mostly salps (tunicates) as expected; Baxter's dogfish eat mostly fish, crustaceans, molluscs, and tunicates as expected; and invertebrate herbivores (kina and paua) eat mostly algae, although they should also eat some phytoplankton, which they do but it is lost in the detail.

## CALIBRATION

Calibration of the model included ensuring stable biomass trajectories when applying no fishing; realistic realised diets; realistic growth and mortality (size-at-age and proportions-at-age); and biomass decreasing with increasing trophic level following the PREBAL (*Link, 2010*) guidelines.

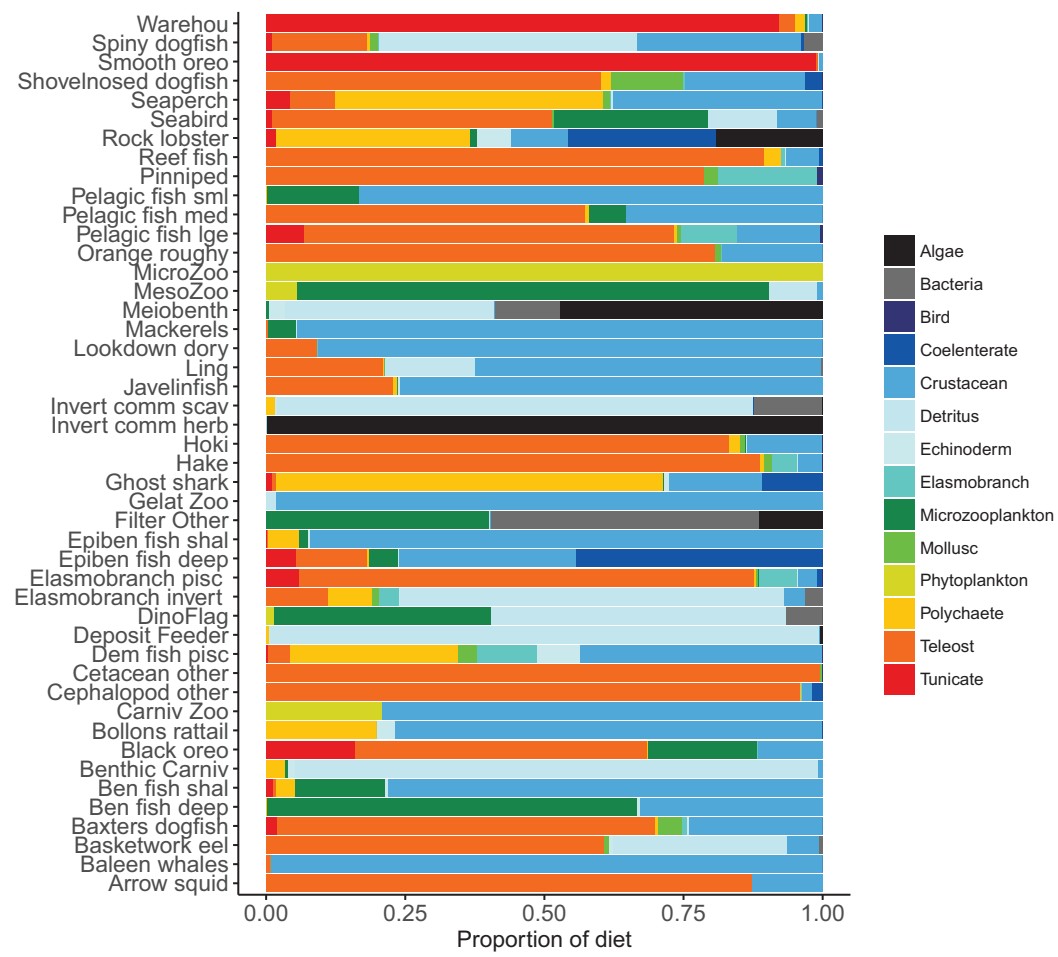

**Figure 5 Summary of the proportion of prey groups in the diets of species functional groups (Tables 2 and 3) over model years 1900–2016 from the fished model where the proportion is by mg N consumed.**

Biomass trajectories should reach a quasi-equilibrium when modelled with constant oceanography and no fishing (*Kaplan & Marshall, 2016*). While oceanography is not constant in our non-fishing model as it changes by year (Section: Oceanography), most of the age-structured groups should still be fairly stable. This was generally the case; all biomass trajectories remained within CVs of 20% over the simulated 1900–2016 model period, except for invertebrate scavengers (commercial) and seaperch. Invertebrate scavengers (commercial) are primarily scampi, and they are likely responding to changes resulting from the oceanographic variables. Biomass trajectories for all age-structured groups from the un-fished model are in Fig. A1. Seaperch biomass was trending downward initially, but they seem to have reached an equilibrium by about 1950, with expected growth and mortality rates.

Atlantis simulates growth rates of age-structured groups as a function of consumption. If growth is too slow, this may be due to insufficient food available, the feeding search rate could be too low or handling time too high, and the reverse of these when growth is too fast. Simulated growth rates of age-structured species groups were assessed by comparing

the simulated size-at-age with those expected based on growth curve estimates from the literature (Table 6). The overlaid simulated and 'observed' figures were generally very similar (Fig. A2). For each species group, we estimated CVs required to satisfy the hypothesis that the modelled size-at-age were not significantly different from the 'observed' with probability of 0.95. The required CVs were all less than 30% except for epibenthic fish (deep and shallow), invertebrate herbivore (commercial), invertebrate scavenger (commercial), ling, rock lobster, and small pelagic fishes. For all these groups, the first age class, and sometimes the first few, were larger in size than expected. Deep epibenthic fish were larger than expected at all age classes, but for all other groups the characteristic of larger than expected size at age had been remedied by the time they were adults.

Natural mortality in the model consists of mortality intrinsic within the model from predation, starvation, and light, oxygen or nutrient deprivation, and additional forced mortality. The latter was applied for modelled species groups that would not otherwise suffer sufficient natural mortality within the model, such as those that have little known predation. Age-structured simulated natural mortality rates from the stable base model were compared to estimates of $M$ from the literature where available (Table 6) by comparing the proportions-at-age. The overlaid simulated and 'observed' figures were generally very similar (Fig. A3), although rock lobster and invertebrate herbivore commercial (primarily paua and kina) had slightly more mortality in the model, and demersal piscivores, epibenthic fish small, pelagic fish medium, and warehou had slightly less mortality.

We summarised biomass by trophic level for the base model from 1900–2016 on a log-scale, and biomass reduced with increasing trophic level with a fitted slope of −1.5 (Fig. 6). This was close to the recommended range of PREBAL of (−1.5, −0.5). The biomass at trophic level 4 was slightly higher in this summary than in the model, as the summary was based on adult trophic level and many of the fish species are trophic level 4 as adults, but lower as juveniles. This resulted in the biomass of the juveniles for these fish adding to the level 4 biomass whereas in the model they were perhaps functioning as a level 3.

# SENSITIVITY ANALYSES

## Oceanography

Oceanographic variables from a ROMS model for years 1996–2004 were used to define temperature, salinity, and flux (water exchange). As our model spanned more than these years, we needed to recycle the ROMS variables in some way. The purpose of this section has two parts: (1) establishing confidence intervals for our model simulations with respect to oceanographic variability; (2) assessing the effect of repeating oceanographic variables from any one year, and whether these take the model outside of the established confidence intervals.

To retain realistic within-year dynamics, the ROMS variables from each year were kept together as a unit, and the years covered by the ROMS model were considered the samples. We ran two sets of simulations: the first sampled ROMS years at random with replacement for each model year simulated (bootstrapped the ROMS years) and repeated this for 50 model runs; the second repeated one ROMS year for all model years simulated and did a separate model run for each of the nine ROMS years. In both cases,

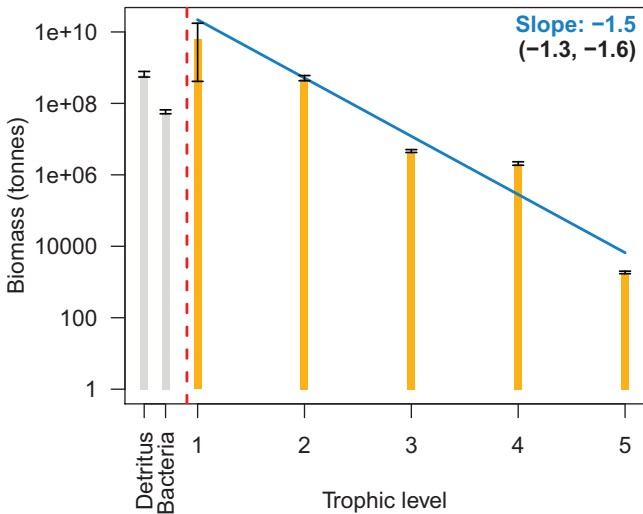

**Figure 6 Biomass by trophic level with 95% confidence intervals from the 1900 to 2016 Chatham Rise Atlantis model simulation.** The blue line is the fitted linear model to the median biomasses by trophic level, the slope which is in blue. The slopes of the linear models fitted to the upper and lower 95% confidence interval limits are given in brackets.

the 2003 ROMS was repeated for a 35-year burn-in period, followed by a 50-year simulation. The 2003 ROMS was chosen for the burn-in period as this year had the closest sea temperatures to the means from all ROMS years (Fig. 7). Bootstrapping the ROMS years was used to establish confidence intervals with respect to between-year oceanographic variability. Repeating each ROMS year in turn was testing the effect of multiple years being different to the other years in some consistent way, such as cooler or warmer.

The established biomass confidence intervals were fairly narrow for most species groups, with CVs <10%. Of the exceptions, diatoms had the highest CV of 79%, followed by carnivorous zooplankton (46%), labile detritus (23%), sediment bacteria (13%), invertebrate scavengers (commercial) (12%), refractory detritus (12%), meso-zooplankton (11%), and pelagic bacteria (11%). That these groups were found to be most sensitive to oceanographic variability in the model is a plausible and sensible result.

The years with cooler sea temperatures (1996, 1997, and 2004) when repeated for 50 years produced the most species groups that went above the established biomass confidence intervals, with the on average warmer years (1999, 2000, and 2001) having the most species groups that went below (Fig. 8). These species groups affected by warmer or cooler years had quite a bit of overlap, with meso-zooplankton, meiobenthos, and black oreo most often affected. All of the species groups that went lower in warm years also went higher in cool years. The reverse was not true; three species groups (arrow squid, labile detritus, and ghost shark) went higher in the cool years, but not lower in the warm years.

Years 2003 and 1998 were closest to the average sea temperatures and had the least number of species groups outside the bootstrap confidence intervals. The Base Model that repeated the ROMS from all nine years in order for the entire model simulation had 16 species groups that exceeded the bounds at some point (less than the warm years) and six species groups that went below the bounds at some point (less than the cool years) (Fig. 8).

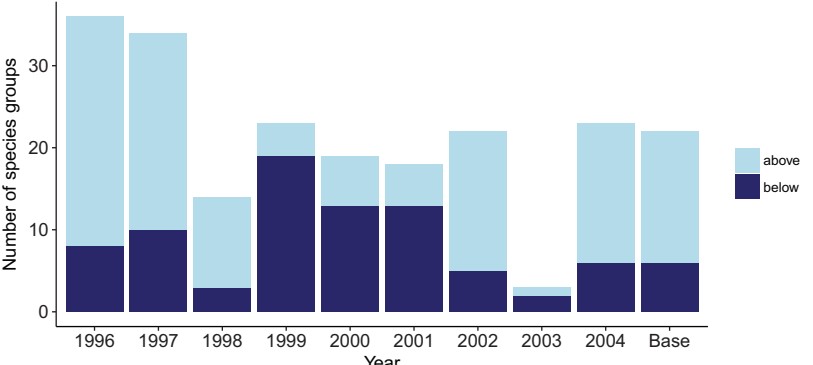

**Figure 7** Sea temperature (°C) from ROMS model outputs by day for each year 1996–2004 (dark blue line) and median sea temperature over all ROMS model years 1996–2004 (grey line).

**Figure 8** Number of species groups for each simulation with one ROMS year repeated that went above (light blue) or below (midnight blue) the limits of bootstrapped ROMS simulations and for the Base Model where the years were repeated in order for the entire model simulation.

## Connectivity and influence

Understanding which species groups are most influential or responsive in the model is another test for realistic dynamics, and may be useful to help understand results of

scenarios explored using this model in the future. To do this, we need to perturb each species group in turn, then assess the responses of the other groups in the system. For each age-structured species group, we ran two simulations, one with a small additional mortality and one larger; $M$(per year) + (0.1, 0.005). We assessed responses of the groups with respect to the Base Model at the completion of 50-year simulations. We analysed the 'keystoneness' and responsiveness of the groups based on biomasses relative to the Base Model.

We calculated keystoneness using an adaption of the method in *Libralato, Christensen & Pauly (2006)*. It is a measure of the effect the group has on the rest of the system (change in biomass of the other species groups), that takes into account its proportion of the total biomass. For example, if two species groups have the same effect, but one has a large biomass and one a small biomass, the smaller would have a larger keystoneness. We used simulation outputs to estimate the total effect ($\varepsilon$) of each species group (Eq. (1)) which used the change in biomass of each group relative to the Base Model (Eq. (2)). The simulated change in biomasses ($S_{f,g}$) were used in place of the mixed trophic impact values calculated from mass balanced models and used by *Libralato, Christensen & Pauly (2006)*. As the additional mortality applied in our simulations caused larger and smaller changes to the focus groups, we scaled the focus groups' biomass proportions by their change in biomass ($S_{f,f}$ in Eq. (4)). Hence, the resulting keystoneness allowed for the effect changing each group had on the other groups, the focus groups biomass as a proportion of the total, and the proportional change in biomass of the focus group relative to the base model.

$$\varepsilon_f = \sqrt{\sum_{g \neq f}^{G} S_{f,g}^2} \tag{1}$$

$$S_{f,g} = \frac{B_{f,g} - B_{b,g}}{B_{b,g}} \tag{2}$$

$$\kappa_f = log(\varepsilon_f(1 - p_f)) \tag{3}$$

$$p_f = \frac{B_{b,f}}{\sum_{g=1}^{G} B_{b,g}} \times |S_{f,f}| \tag{4}$$

$\varepsilon_f$, effect group $f$ has on the other groups

$S_{f,g}$, proportional change in biomass of group $g$ when group $f$ was reduced, relative to the Base Model

$B_{b,g}$, $B_{b,f}$ biomass in base model of group $g$, $f$

$B_{f,g}$, biomass of group $g$ in model with group $f$ mortality increased

$\kappa_f$, keystoneness of group $f$

$p_f$, biomass proportion of group $f$

There were four species groups that stood out as having more effect than the other groups: orange roughy, hoki, pelagic fish small (primarily myctophids), and spiny dogfish. These remain the top four for keystoneness, but the order changes due to the proportional biomasses (Fig. 9).

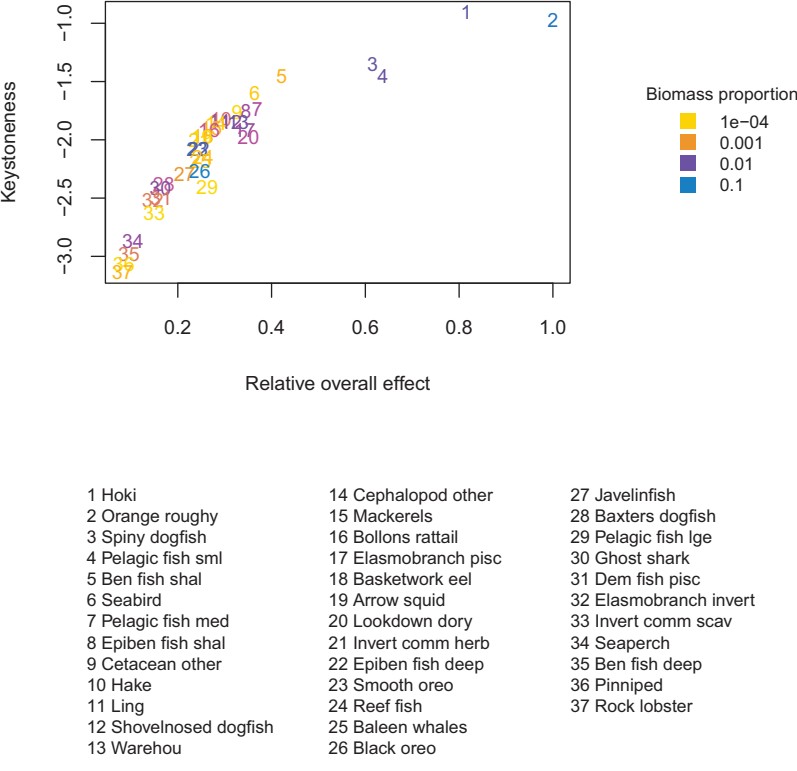

1 Hoki
2 Orange roughy
3 Spiny dogfish
4 Pelagic fish sml
5 Ben fish shal
6 Seabird
7 Pelagic fish med
8 Epiben fish shal
9 Cetacean other
10 Hake
11 Ling
12 Shovelnosed dogfish
13 Warehou

14 Cephalopod other
15 Mackerels
16 Bollons rattail
17 Elasmobranch pisc
18 Basketwork eel
19 Arrow squid
20 Lookdown dory
21 Invert comm herb
22 Epiben fish deep
23 Smooth oreo
24 Reef fish
25 Baleen whales
26 Black oreo

27 Javelinfish
28 Baxters dogfish
29 Pelagic fish lge
30 Ghost shark
31 Dem fish pisc
32 Elasmobranch invert
33 Invert comm scav
34 Seaperch
35 Ben fish deep
36 Pinniped
37 Rock lobster

**Figure 9 Keystoneness (*y*-axis) and relative overall effect (*x*-axis) for all age-structured species groups, with numbers giving keystoneness ranking (1 is the most influential using Eq. (3)).** Colours indicate biomass proportion scaled by proportional change in biomass (Eq. (4)).

We calculated responsiveness in a similar way to keystoneness, but from the perspective of the response group (Eq. (5)).

$$R_g = \sqrt{\sum_{f \neq g}^{G} \left( m_{f.g}^2 \times p_f \right)} \tag{5}$$

$R_g$ responsiveness of group $g$ to increased mortality in all other groups

The most responsive group was pelagic fish small (primarily myctophids), followed by smooth oreo, invertebrate scavengers commercial (primary scampi), and pelagic fish medium (primarily barracouta) (Fig. 10). The pelagic fish small species group ranked high for keystoneness and responsiveness, and so may be most important and influential in scenarios explored with this model.

## FISHING

Most of the fisheries on the Chatham Rise became established after the mid-1970s, with the exception of the blue cod (*Parapercis colias*) (reef fish species group) fishery which extends back to the early 1900s. Individual catch histories are in Fig. A4 and Fig. 11 presents a summary of catches from the Chatham Rise with the top six species by total catch shown in colour and the others combined into an 'other' category. Hoki had the largest total
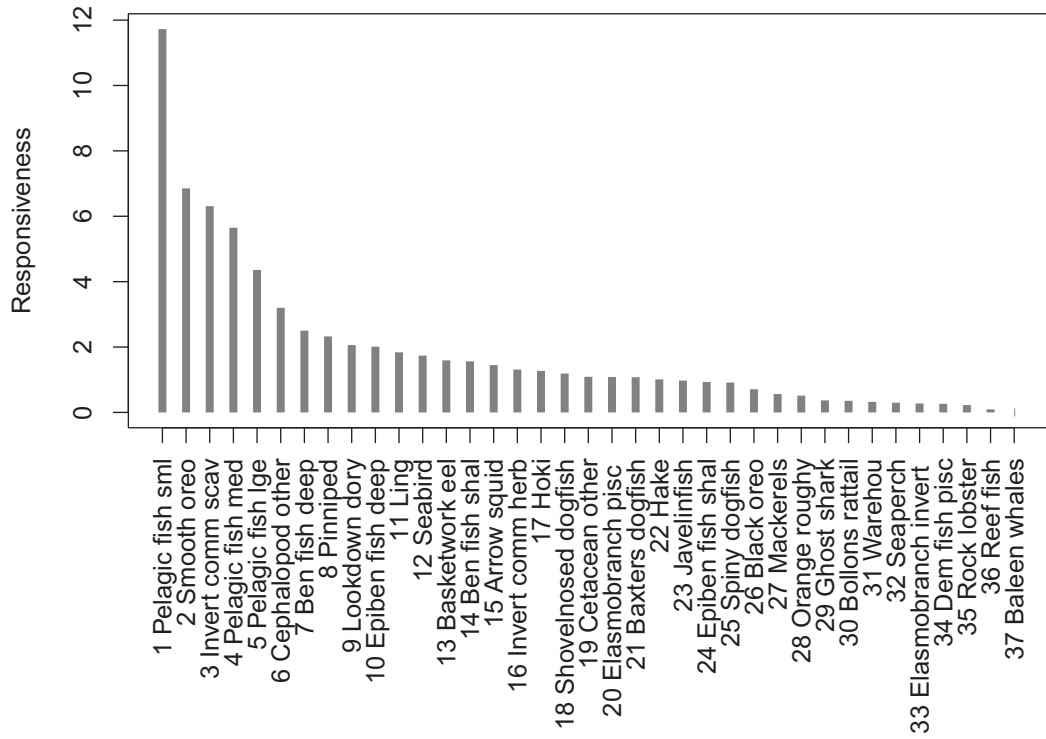

**Figure 10** Responsiveness of age-structured species groups after 50 years of perturbation, as calculated in Eq. (5).

catch, followed by orange roughy, smooth oreo, ling, black oreo, then barracouta. Orange roughy comprised the largest individual fishery in the late-1970s–early-1990s after which it declined markedly; from the 1990s hoki was the dominant fishery.

The fisheries were modelled with six fleets, defined in Table 7. The demersal line fishery was dominant until mid–late 1960s when the demersal trawl fishery became dominant, catching approximately 70,000 tonnes per year (Fig. 12). The historical catches from these fleets were forced in the model using spatially and temporally resolved inputs.

## Comparison with fisheries CPUE and stock assessment indices

CRAM model estimates of biomass trends for key fisheries species were compared to CPUE and/or stock assessment indices where these were available. The Atlantis model captures the main biomass trends of hoki in response to historical fishing (Fig. 13). Hoki are the largest fishery on the Chatham Rise, and has one of the most complex stock assessment models in New Zealand, with multiple areas, intricately defined migration, and annual recruitment deviates (*McKenzie, 2016*). The Atlantis model results are very similar to the stock assessment model results for hake and ling, and although the stock assessment models for these are not as complicated as hoki, they still have between-year recruitment deviates (*Horn, 2013*; *McGregor, 2015*) that are not present in the Atlantis model. The species group 'Invertebrate scavengers (commercial)' is primarily scampi, and the matched increase in the late 1990s–early 2000s is particularly pleasing as catches were fairly constant over this time (*Tuck, 2016*), so the increase

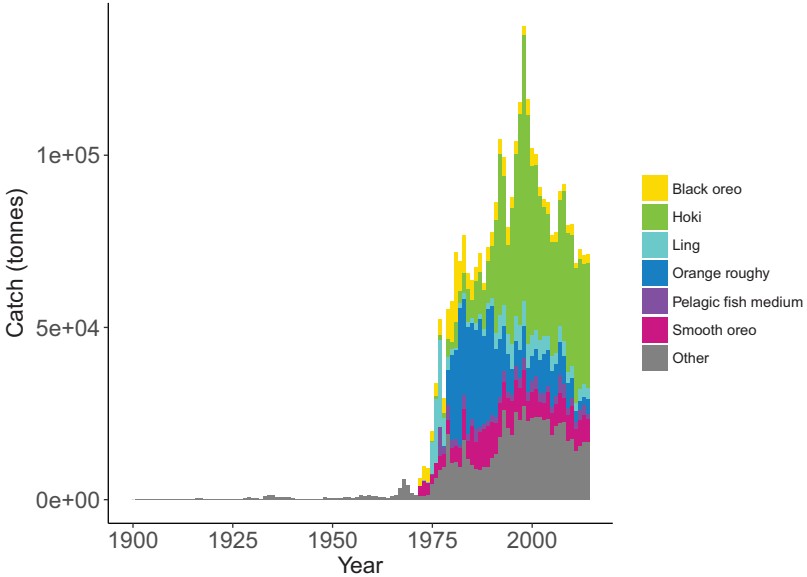

**Figure 11** Tonnes caught from Chatham Rise 1900–2014 for all species with top six species groups by total catch coloured separately.

**Table 7 Fishing fleets defined for Chatham Rise Atlantis model.**

| Code | Description | Number of species groups | Total catch (t) |
|------|-------------|--------------------------|-----------------|
| trawlDEM | trawl on demersals and mesopelagics | 33 | 2,850,000 |
| lineDEM | line on demersals and mesopelagics | 16 | 1,200,000 |
| snetDEM | setnet on demersals and sharks | 6 | 45,700 |
| potIVS | potting on lobster and blue cod | 4 | 241,000 |
| jigCEP | jig on squid | 1 | 1,700 |
| diveIVH | diving on paua and kina | 2 | 158,000 |

**Note:**
Number of species groups is the number of species groups that have been caught by each fishing fleet; total catch is the total tonnes caught by each fishing fleet from 1900 to 2014.

is coming from dynamics within the model. Orange roughy is a close match to the stock assessment, even though this stock assessment model also has between-year recruitment deviates (*Dunn & Doonan, in press*) that are not in the Atlantis model. The magnitude of the stock assessment biomasses (unscaled) are compared to the CRAM biomasses in the inset boxplots in Fig. 13. Hoki, hake, and invertebrate scavengers (commercial) were all close to one, indicating matched magnitudes between the stock assessment and CRAM biomasses. Ling were generally less than one, indicating the CRAM biomasses were larger than the stock assessment biomasses. Orange roughy were greater than one, indicating CRAM biomasses were smaller than the stock assessment biomasses.

## SKILL ASSESSMENT

Quantitative skill assessments have become popular as part of assessing the performance of Atlantis models (*Sturludottir et al., 2018*; *Ortega-Cisneros, Cochrane & Fulton, 2017*;

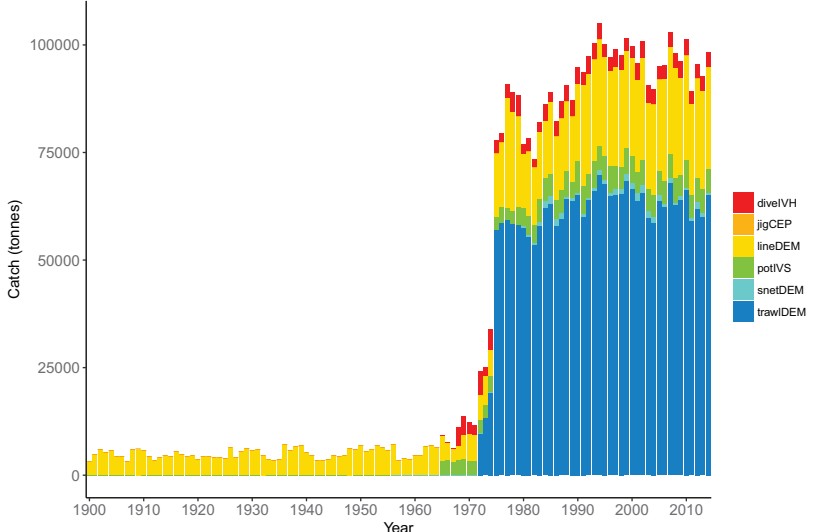

**Figure 12 Total tonnes caught by fishing fleet from the Chatham Rise 1900–2014.** Descriptions for the fleet codes are in Table 7.

*Olsen et al., 2016*). A quantitative skill assessment was carried out, comparing model biomass estimates with those from trawl surveys where available (*O'Driscoll et al., 2011*; *Stevens et al., 2017*). The trawl surveys target hoki, hake, and ling, and as such the biomass indices are most reliable for these three species. The metrics selected were three of those suggested in *Olsen et al. (2016)* and *Stow et al. (2009)*: modelling efficiency (MEF) used to asses model predictions relative to the mean of the observations (Eq. (6)); reliability index (RI) gives the average factor the model predictions differ from observations (Eq. (7)); Pearson's correlation (*r*) assesses whether model predictions are correlated with observations (Eq. (8)). The full set of CRAM biomass trajectories with historic catches and trawl survey indices are in Fig. A4.

$$MEF = \frac{\sum_{y=1}^{Y}\left(O_y - \bar{O}\right)^2 - \sum_{y=1}^{Y}\left(O_y - P_y\right)^2}{\sum_{y=1}^{Y}\left(O_y - \bar{O}\right)^2} \tag{6}$$

$$RI = exp\sqrt{\frac{1}{Y}\sum_{y=1}^{Y}\left(\log\frac{O_y}{P_y}\right)^2} \tag{7}$$

$$r = \frac{\sum_{y=1}^{Y}\left(O_y - \bar{O}\right)\left(P_y - \bar{P}\right)}{\sqrt{\sum_{y=1}^{Y}\left(O_y - \bar{O}\right)^2 \sum_{y=1}^{Y}\left(P_y - \bar{P}\right)^2}} \tag{8}$$

where
$Y$ is the number of years for which there are observations,
$O_y$ is the observed biomass in year $y$,
$P_y$ is the model biomass in year $y$

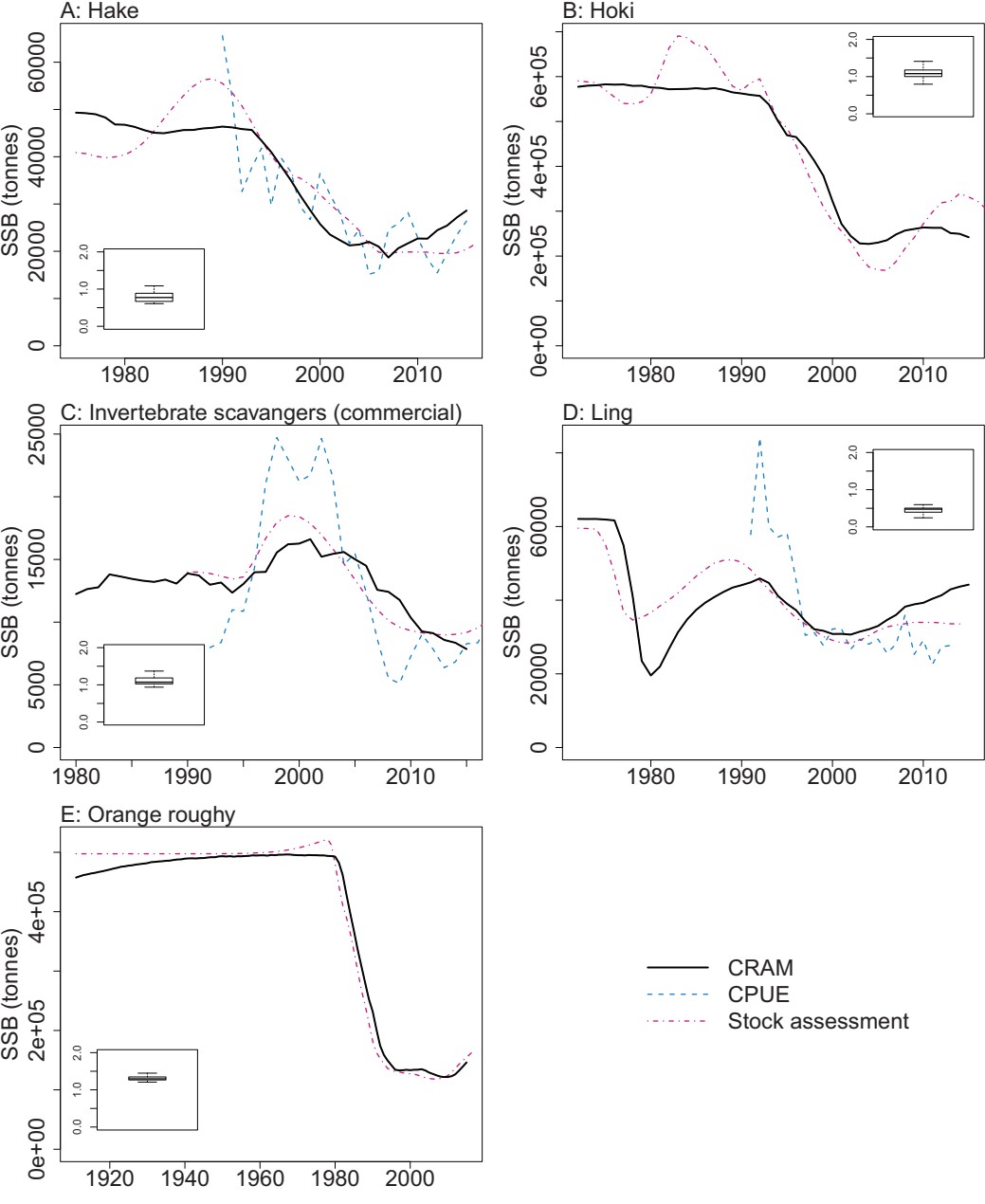

**Figure 13** CRAM estimated spawning stock biomass (SSB) (black solid), stock assessment estimated SSB (red dot-dash), and CPUE (blue dash) where available for the hake (A), hoki (B), invertebrate scavengers (commercial) (primarily scampi) (C), ling (D), and orange roughy (E). CPUE and stock assessment SSB were rescaled to match the mean of the CRAM estimated SSB. Inset boxplots show the range of values for the corresponding unscaled stock assessment SSB divided by the CRAM estimated SSB.

Each skill assessment metric was calculated using single point estimates from the trawl survey, and variants on RI and MEF were calculated allowing for the trawl survey estimated 95% confidence intervals. Both variants only penalised the skill metric for terms outside of the 95% confidence intervals of the trawl survey.

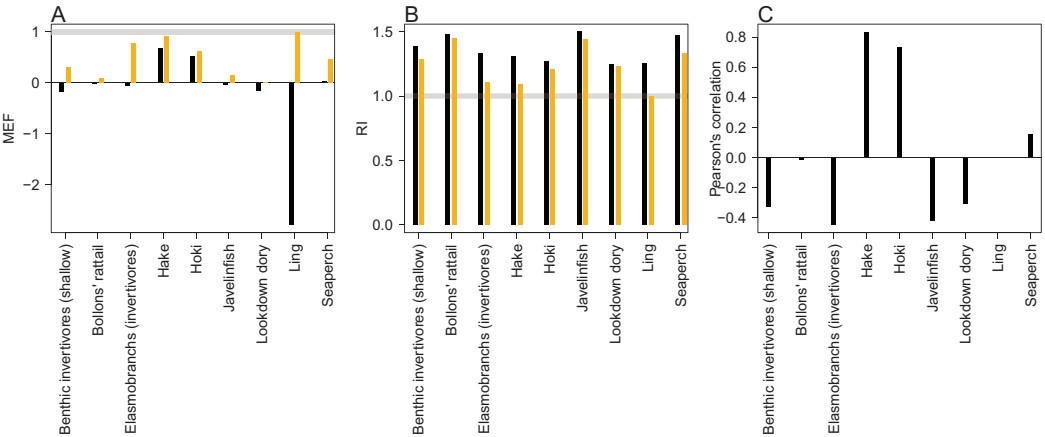

**Figure 14 Skill assessment metrics MEF (A), RI (B), and Pearson's correlation (C) for CRAM species groups that have trawl survey indices for abundance.** Metric definitions in Eqs. (6–8). The black bars are the skill metrics with respect to single point estimates from the trawl survey. The orange bars are the skill metrics with respect to the trawl survey 95% confidence intervals. The grey horizontal lines in the MEF and RI figures mark the value for a perfect fit, which is 1 for both of these.

An MEF close to one indicates a close match between model predictions and observations, with zero indicating the mean of the observations is as close as the model predictions, and a negative value indicating the model predictions fit the observations worse than the mean of the observations. When the observed values are roughly stationary about the mean, as was the case for ling, it is difficult for the predictions to improve on the mean of the observations. Ling stands out at approximately −2.5 when compared to the trawl survey point estimates, but as all the predicted points for ling sit within the 95% confidence interval, it receives a score of one when taking the bounds into account (Fig. 14). Benthic invertivores (shallow) and lookdown dory are slightly negative with respect to the trawl survey point estimates.

A RI of one indicates the model predictions are exactly equal to the observations. RI greater than one (it cannot be less than one) indicates the factor by which observations are on average different to predictions. Since $\log(O/P)$ is equal to $-\log(P/O)$ and the RI squares these terms, an observation, that is, half the prediction will contribute exactly the same to this index as an observation, that is, twice a prediction. Hence, a RI of 2 indicates the observations differ from the predictions on average by 2, but these could be generally twice as big or half as big, or both. All groups had RIs between 1 and 1.5 (Fig. 14), indicating the observations are at worse on average 1.5× the predictions or (2/3)× the predictions.

A Pearson's correlation close to one indicates trends in the predictions vary with those in the observations, close to zero indicates there is little relationship between the trends, and negative indicates the predicted trends tend to be opposite from the observed trends. Hake and hoki had good correlation, close to 0.8. The other groups were either close to zero or negative (Fig. 14). This is neither surprising nor concerning as the trawl survey estimates for these groups tend to have high variability and high CVs which are not taken into account here.

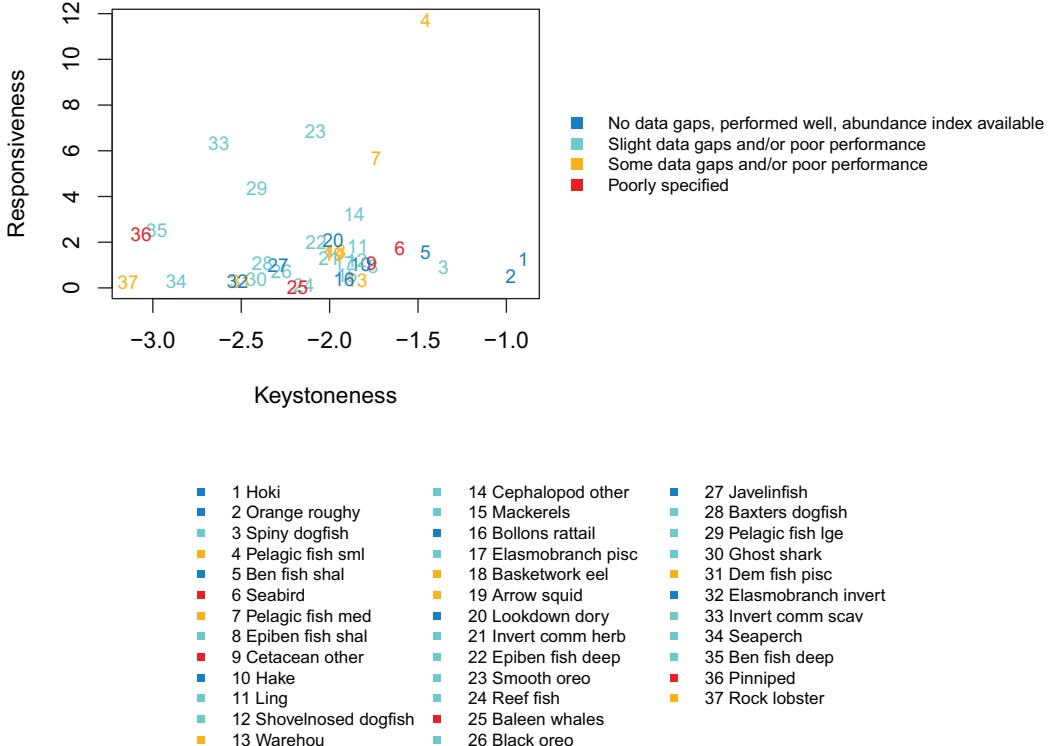

**Figure 15 Keystoneness (*x*-axis) and responsiveness (*y*-axis) with numbers showing keystoneness ranking and colours how well each species group was informed and/or performed in the model (legend).**

# BRINGING IT TOGETHER

We qualitatively graded the species groups by how well they performed in the model and how well informed they were by data, information and other research (referred to as 'informance'). We compared these gradings with the keystone and responsiveness from 'Section: Connectivity and influence'. Figure 15 gives a visual guide for how well the most influential or responsive species groups did for informance and performance. While poor knowledge may not be concerning if paired with high responsiveness providing keystoneness is low (since the effects may be more limited to this single species group), the triple of highly responsive, a keystone species, and poorly defined may need consideration for future scenarios.

The groups that were highest for keystoneness and highest for informance and performance were hoki, orange roughy, benthic fish shallow (primarily oblique banded rattail), and hake. These all have abundance indices available, biological parameters, diet information, and all perform well with respect to these in the model. Hoki, orange roughy and hake (groups 1, 2, and 10 for keystoneness) have full stock assessments, which the model matches well. These are important groups for fisheries and will likely feature strongly in any fisheries scenarios explored with this model.

Species groups Pelagic fish small (primarily myctophids) and Pelagic fish medium (primarily barracouta) were both high with respect to keystoneness and responsiveness,

and while both were fairly well defined, these had some areas of poor model performance and do not have abundance indices to compare. The estimated length at age 1 from CRAM for small pelagic fish is larger than expected. This may be due to the size of recruits being larger than they should be, or the fish eating (and hence growing) more than they should in this first year. They are not so big that the effect transfers to the age-2's, as the age-2's are the correct size (Fig. A2), so this is probably not influential on the model overall. Medium pelagics have slightly less natural mortality in the model than they should (Fig. A3), and may be less responsive to fishing mortality as a result. As they are seventh with respect to keystoneness and high for responsiveness, they could affect scenario outcomes and are worth considering when analysing results. They make up approximately 1% of the age-structured biomass.

Spiny dogfish were third for keystoneness, and low for responsiveness. They fit well to mortality and growth curves, but we do not have an index of abundance with which to compare the model simulated biomass in response to historical fishing. They make up approximately 5% of the age-structured biomass.

Epibenthic fish shallow (primarily common roughy) were eighth for keystoneness, but low for responsiveness. They compare reasonably well to the trawl survey abundance index, but have less natural mortality in the model than they should. They make up approximately 1% of the age-structured biomass.

Species groups 'Seabird' and 'Cetacean other' are both poorly defined and rank within the top 10 for keystoneness, although lower for responsiveness. They are both composite groups, with Seabird consisting of all sea and shore birds, and Cetacean other consisting primarily of sperm whales, pilot whales and dolphins (Table 2). Scenarios explored in the future may benefit from sensitivity analysis with respect to these two groups to understand their effect on the outcomes, or perhaps some more work to better define them.

## DISCUSSION

Ecosystem-based fisheries management is most likely to be achievable with the best information and modelling available (*Heymans et al., 2010*). The Chatham Rise Atlantis model presented here uses the wealth of data and information available for the Chatham Rise and its fisheries, and one of the best ecosystem models for exploring 'what-if' type questions (*Plagányi, 2007*) and ecosystem-level management strategy evaluation (*Fulton et al., 2014*). This comprehensive ecosystem model with realistic population dynamics and flow-on effects has the potential to be a valuable tool for understanding potential system-wide responses to fisheries management strategies in one of New Zealand's largest fishing grounds.

Some key aspects of this model performed convincingly well, such as responses of key fisheries species under fishing, realised diets, and the keystone rankings. That the key fisheries species results were very similar to the corresponding stock assessment results gives confidence that the model can respond to fishing in a way, that is, realistic, and that the ecosystem effects relative to these species are realistic. The stock assessment models fit data such as proportions at length and biomass indices with the help of between-year recruitment deviates, which are not present in the Chatham Rise Atlantis model. Conversely, the stock assessment models do not have time-varying natural

mortality or growth rates, which are present in the Chatham Rise Atlantis model. As such, both modelling approaches achieve similar results but in very different ways. It is possible that the recruitment deviates in the stock assessments are proxy's for the other ecosystem dynamics that the Atlantis model is able to capture (or vice versa). However, the Atlantis model is too complex to fit comprehensively to data and is entirely deterministic. Hence, the Chatham Rise Atlantis model's ability to achieve the same results as the stock assessment models, that were fitted to data, is the best outcome.

Realistic diets and the influence of species groups on the rest of the ecosystem are key to the model's potential to explore and gain understanding of flow-on and cascading effects. It may be possible, for example, for a species to have realistic growth rates, but it is not very useful in an ecosystem modelling context if they do so by eating the wrong things. While they might respond realistically to direct pressure such as fishing, the flow-on effects would not likely reflect reality. Due to the complex nature of the Atlantis model, the summary of realised diets, together with analysing the keystoneness and responsiveness, are appropriate for determining whether species interactions are generally realistic, at a level of complexity that can be comprehensible. The Chatham Rise Atlantis model has realistic diet summaries for all species groups, and the top keystone species groups were all those we would expect to be most influential within this ecosystem. This is not to say the model could not benefit from further future work examining the realised diets at a finer scale—spatially, temporally, and by age-class.

Exploring the models sensitivity to initial conditions, while not an insignificant amount of work, may be worth doing at some stage in the future to add to our understanding of the models stability and persistence of dynamics. This has not, to the best of our knowledge, been done for Atlantis or OSMOSE models, likely due to the enormous complexity and computing resources required for the task. Sensitivities to initial conditions have been explored using Ecopath (*Essington, 2007*) and Ecopath with Ecosim (EwE) (*Steenbeek et al., 2018*). We are in the early stages of developing an EwE version of CRAM, and it may be more feasible to explore ranges of initial conditions within the EwE framework, with the possibility of then adapting the analyses to the Atlantis model. Sensitivities of high-ranking keystone species, such as spiny dogfish, would be simpler to implement and may produce greater understanding of the model.

While there are some knowledge gaps, we have identified those most likely to influence scenario outcomes through analysing how influential (keystoneness) and influenced (responsiveness) the species groups are on and to each other. The composite groups 'cetacean other' and 'seabirds' were highly influential while poorly specified. Two solutions would be to (a) split these groups into smaller groups that can be better specified; (b) run sensitivities with respect to these groups when exploring scenarios using this model. As option (a) would require more data than we currently have available, option (b) is the only currently viable option.

The oceanographic variables based on years 1996–2004 were found to be influential on the simulated biomasses of the species groups, and the order they were repeated changed the results, with CVs of up to nearly 80%. This suggests scenarios carried out using

this model need to consider oceanographic variability in simulated results, using multiple runs with different oceanographic years repeated or changing the order. This may be true for many ecosystem models, but we are unaware of similar analyses completed elsewhere. Further work understanding which species groups and/or spatial areas of the model are most affected by oceanographic variability might be helpful in understanding potential impacts on scenario results.

As Atlantis is spatially resolved, there is scope for a greater emphasis on the effects of features such as habitats, depth, and oceanographic features on responses to fisheries management scenarios. *Kaplan, Horne & Levin (2012)* explored spatially resolved fisheries management scenarios using an Atlantis model of the California Current, including areas closed to bottom-contact fishing gear, and varying spatial management specification relating to marine protected areas (MPAs). In the Chatham Rise ecosystem, it may be that repeating cooler or warmer years such as carried out in this study could influence the spatial distribution of some species. This could in turn influence the range of plausible responses to fisheries management scenarios that have a spatial aspect, such as MPAs, the effects of different fishing gear, serial depletion of fishing grounds, and potential effects on by catch species that may overlap spatially with species that are targeted by fisheries.

While we have confidence in this model for exploring fisheries type scenarios in support of an ecosystem-based approach to fisheries management, the model still stands to benefit from further exploration. Key to understanding the implications of any results from such a complex model is to first ask what in the model is producing the results, before asking what it tells us about the system.

## CONCLUSIONS

The analyses presented in this paper are intended to set the stage for an understanding of how the model is specified and how it behaves, but it is not exhaustive. The model produces similar results to fisheries stock assessment models for key fisheries species, and the population dynamics and system interactions are realistic. Confidence intervals based on bootstrapping oceanographic variables were fairly narrow for most species groups, with diatoms, carnivorous zooplankton and labile detritus having the largest CVs. The species groups with the highest keystoneness were orange roughy, hoki, pelagic fish small (primarily myctophids), and spiny dogfish. The model components that have knowledge gaps and are most likely to influence model results were oceanographic variables, and the aggregate species groups 'seabird' and 'cetacean other'. We recommend applications of the model include alternatives that vary these components. It is expected that any future use of the model will add first to our understanding of the model, and then possibly to our understanding of the ecosystem.

# APPENDIX

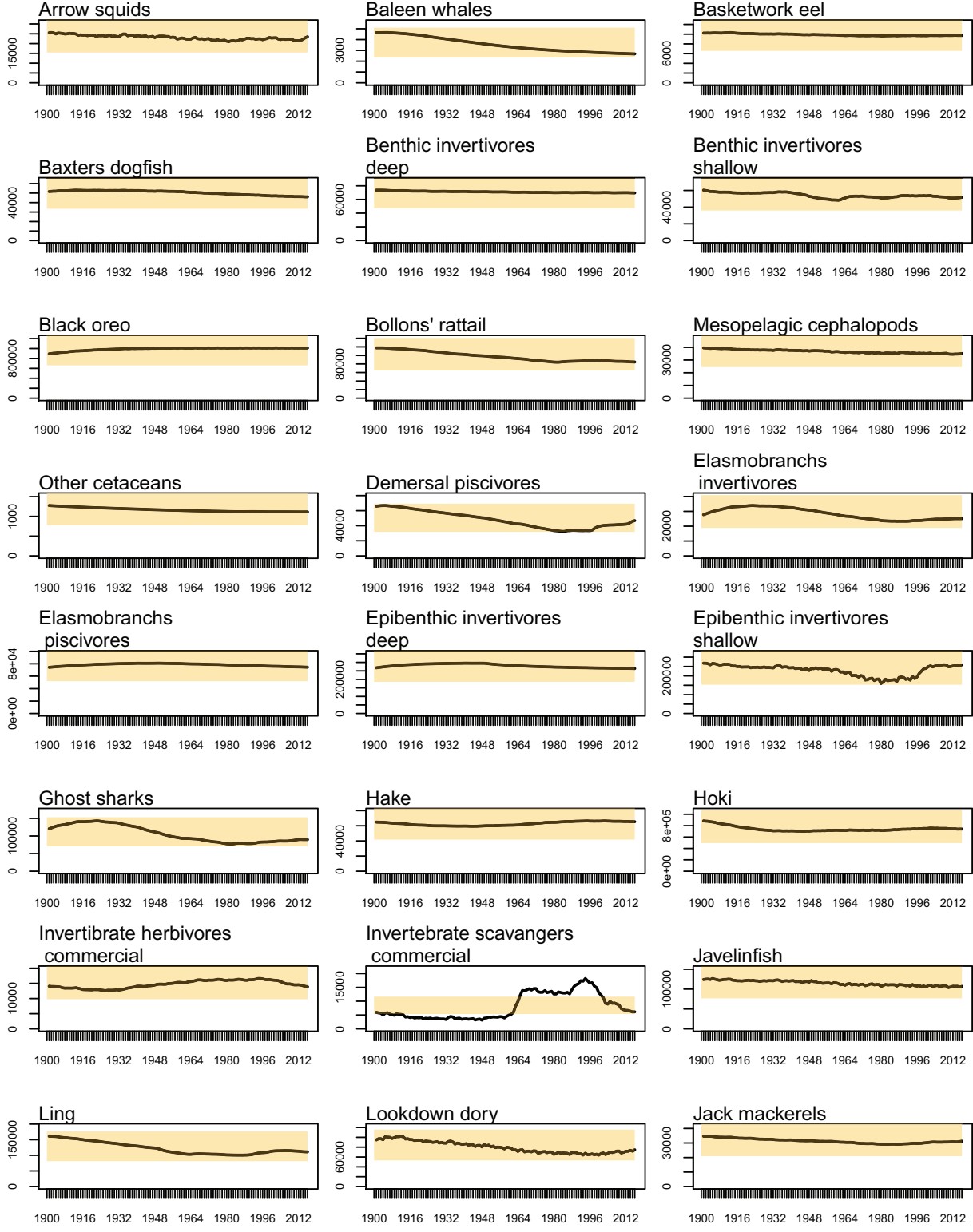

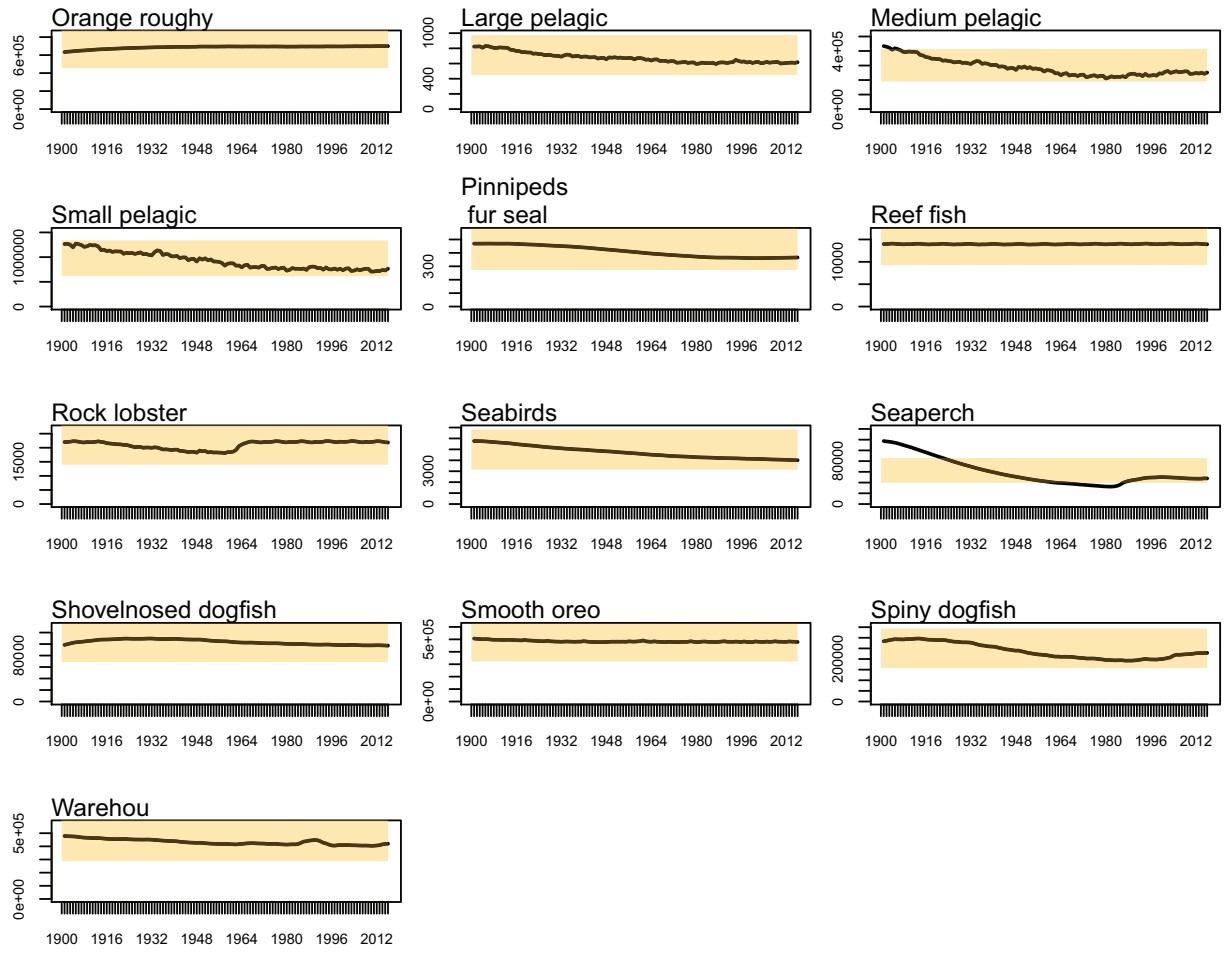

**Figure A1 Simulated biomass from the un-fished model (black line) with 95% confidence intervals based on 20% CVs (Coefficient of Variation) shaded orange by species group.**

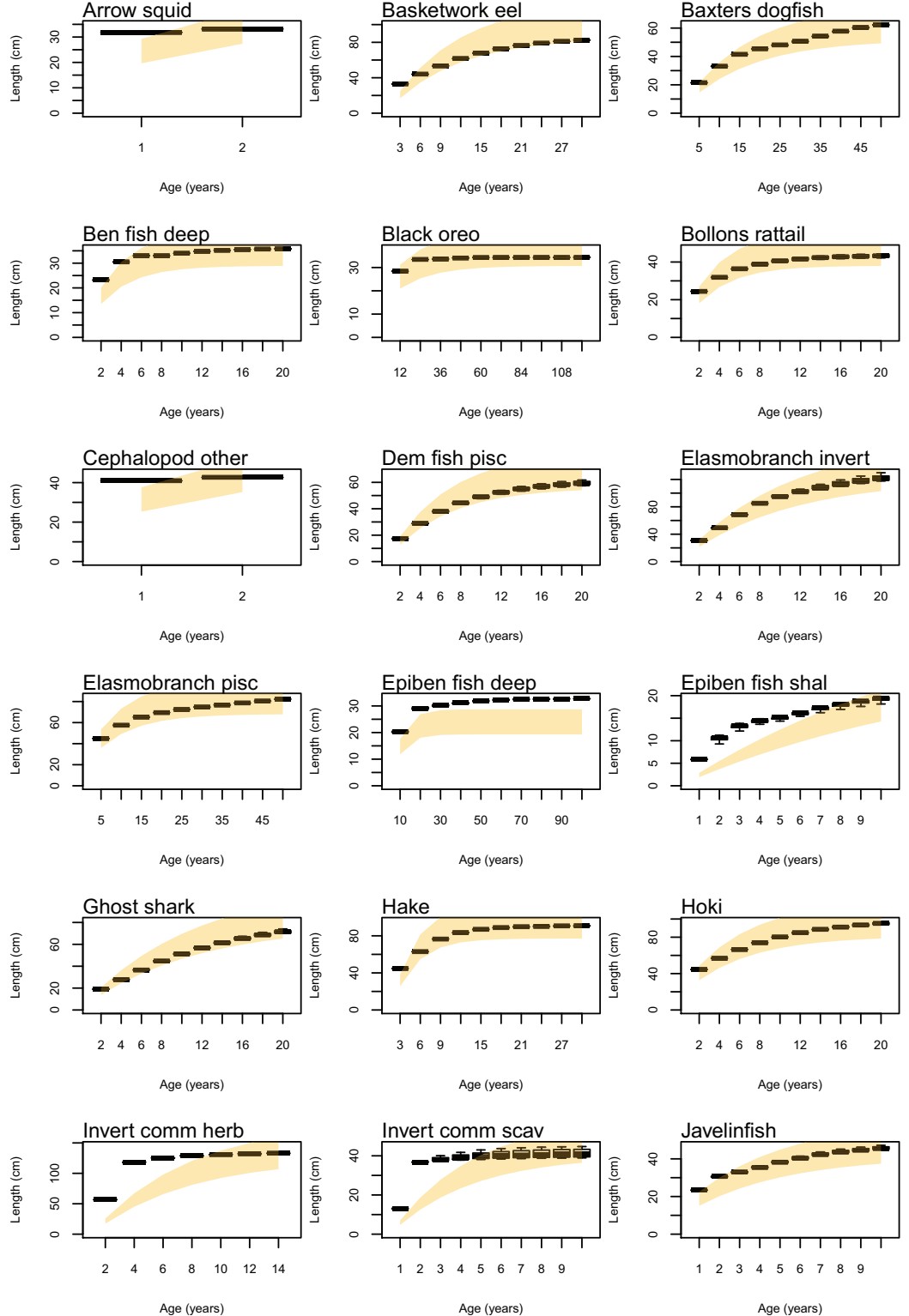

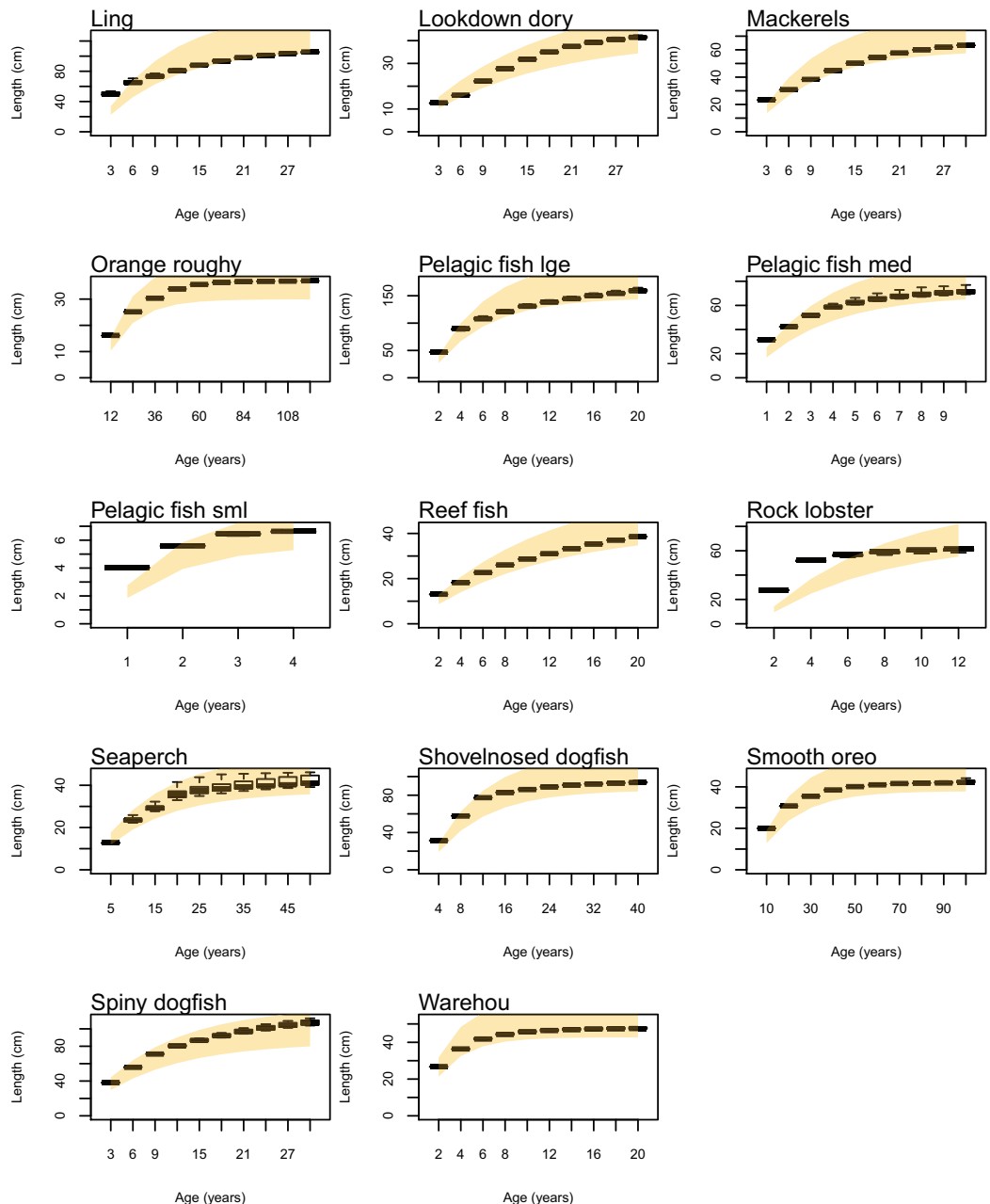

**Figure A2** Size-at-age using values based on literature where available (orange shaded shows 95% confidence intervals using CV 10%) and from CRAM simulated years 1900 to 2016 (boxplots).

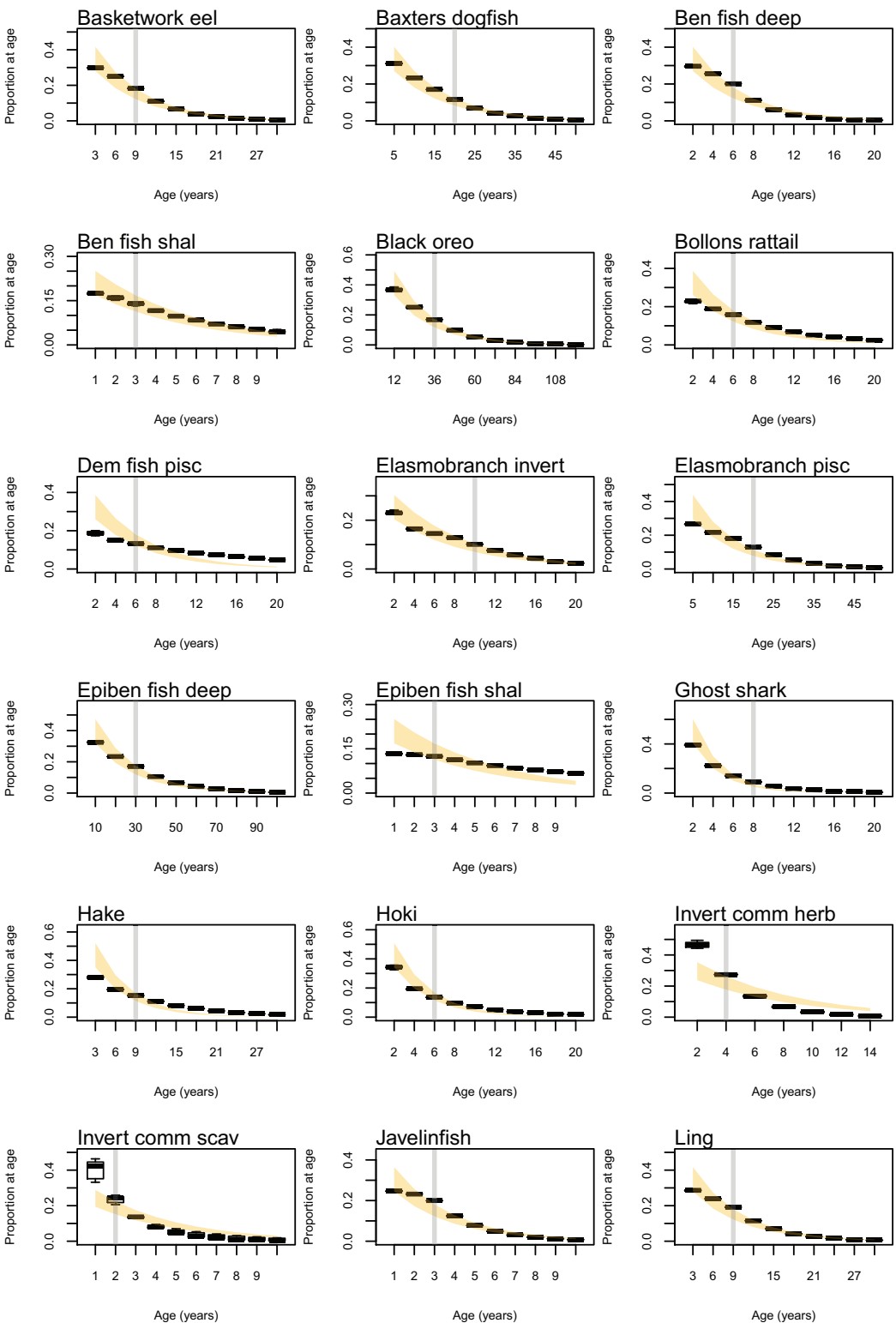

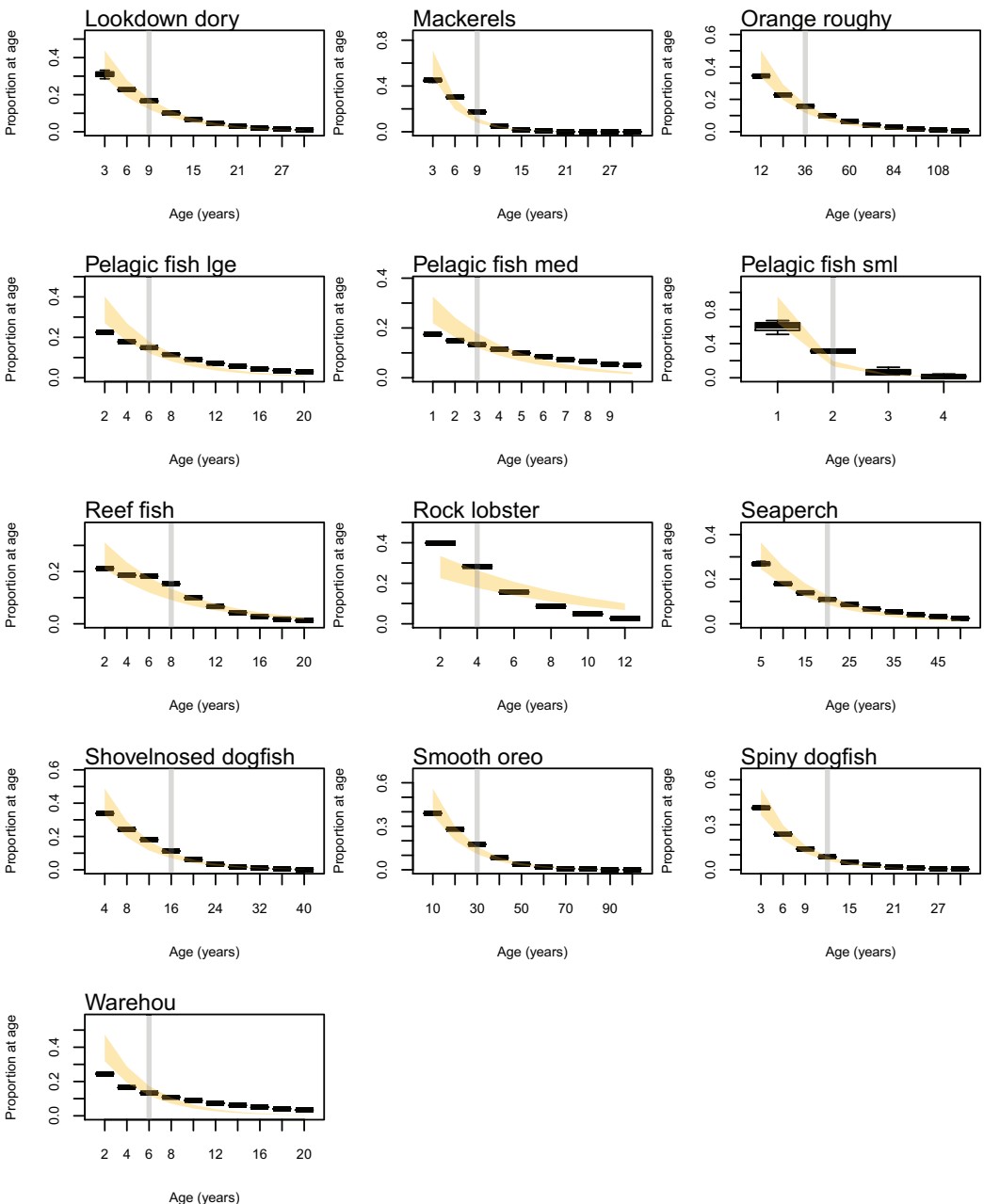

**Figure A3** Proportions at age using M based on literature where available (orange shaded shows 95% confidence intervals using CV 10%) and from CRAM simulated years 1900 to 2016 (boxplots).

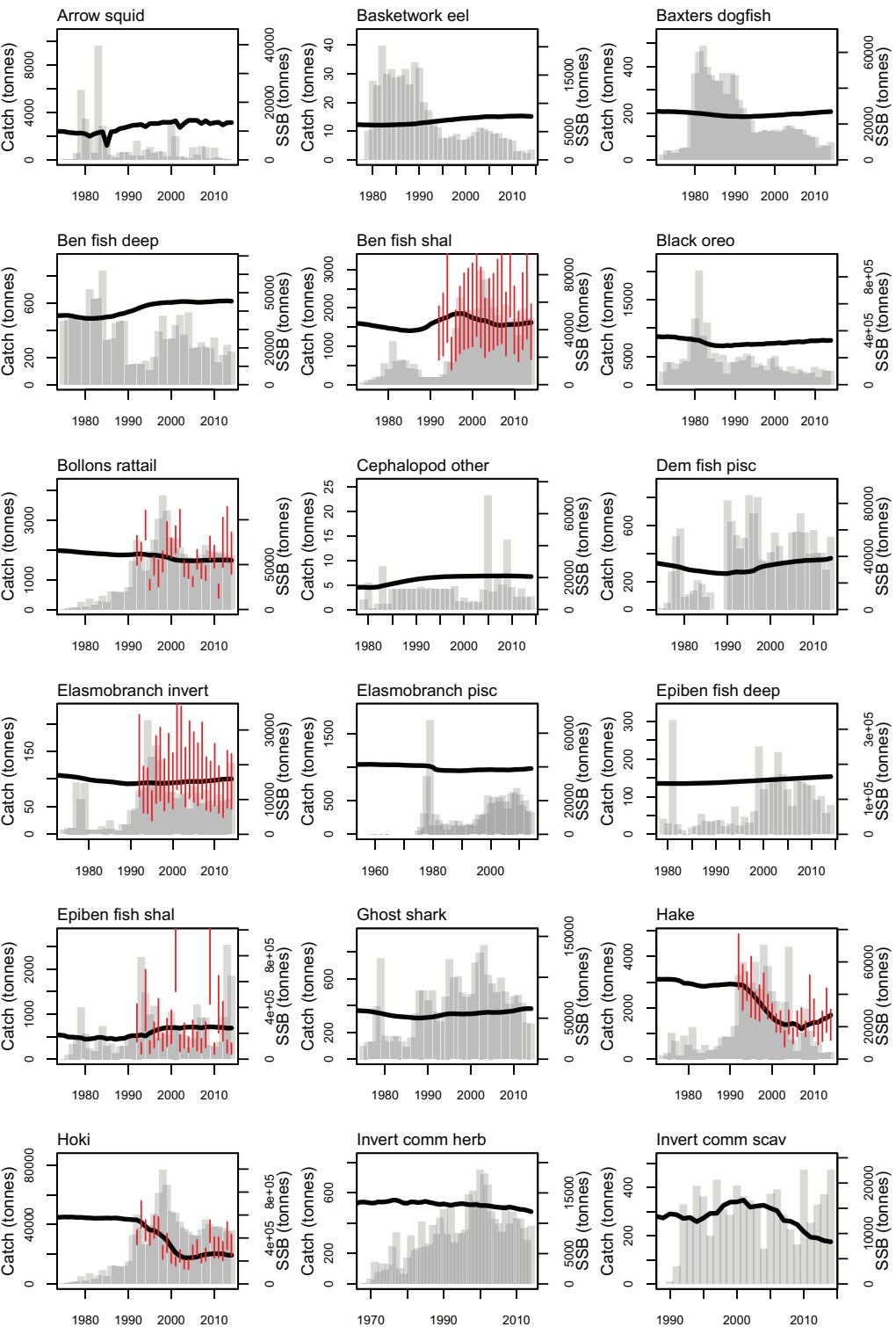

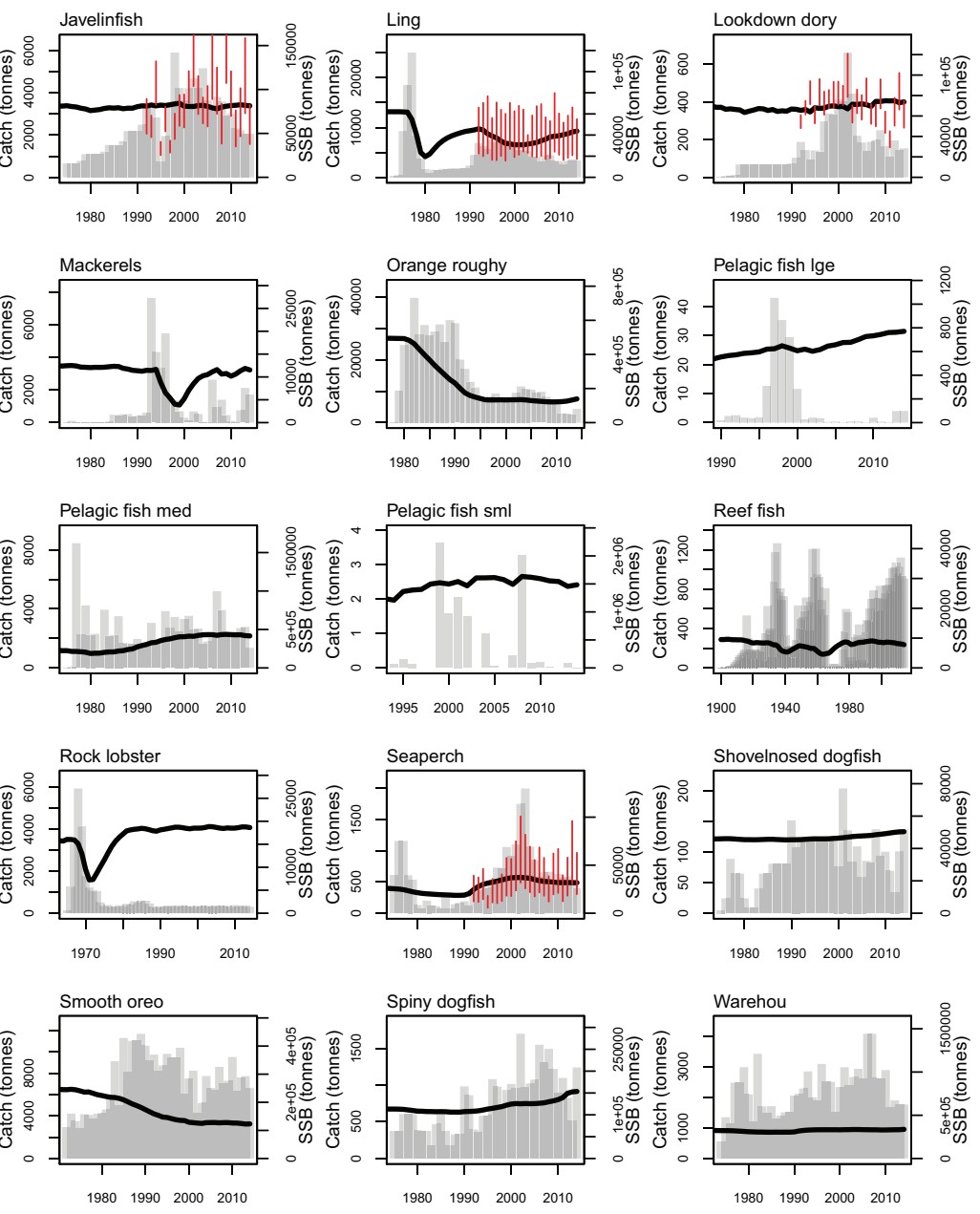

**Figure A4 Observed biomass estimated from trawl surveys (red), estimated biomass from CRAM (black) and forced catch history (grey) for all groups with trawl survey estimates.**

## ACKNOWLEDGEMENTS

Mark Hadfield for development of the ROMS model for oceanographic variables.
Bec Gorton (CSIRO) for converting the ROMS variables into Atlantis model inputs.
Ian Tuck for providing a comprehensive internal review of the manuscript. Cliff Law and Graham Rickard (NIWA) for help with ocean physics data and conversions.
Matt Pinkerton for trophic level results from stable isotope analyses and visible band radiation at sea surface data. James Bell, Victoria University supervisor.

### Funding

This work was funded under NIWA project FIFI1801. The funders had no role in study design, data collection and analysis, decision to publish, or preparation of the manuscript.

### Grant Disclosure

The following grant information was disclosed by the authors:
NIWA project: FIFI1801.

### Competing Interests

Vidette L. McGregor, Peter L. Horn and Matthew R. Dunn are employed by National Institute of Water and Atmospheric Research (NIWA) Ltd. Elisabeth A. Fulton is employed by the Commonwealth Scientific and Industrial Research Organisation (CSIRO).

### Author Contributions

- Vidette L. McGregor conceived and designed the experiments, performed the experiments, analysed the data, contributed reagents/materials/analysis tools, prepared figures and/or tables, authored or reviewed drafts of the paper, approved the final draft.
- Peter L. Horn analysed the data, contributed reagents/materials/analysis tools, prepared figures and/or tables, authored or reviewed drafts of the paper, approved the final draft.
- Elizabeth A. Fulton authored or reviewed drafts of the paper, approved the final draft.
- Matthew R. Dunn authored or reviewed drafts of the paper, approved the final draft.

### Data Availability

Data is available in GitHub: https://github.com/mcgregorv/CRAM.git.

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
