# Peer review of "From data compilation to model validation: a comprehensive analysis of a full deep-sea ecosystem model of the Chatham Rise"

_PeerJ, doi:10.7717/peerj.6517_

## Round 0.1 · original submission · Major Revisions

I am pleased to see this type of work submitted to Peer J. The research effort that is described is substantive and meritorious.

However, the paper offers scarce context on end-to-end ecosystem modeling. This is essential for a paper that claims that “We present the first end-to-end ecosystem model of the Chatham Rise, which is also the first end-to-end ecosystem model of any deep-sea ecosystem.” The Introduction focuses too much and too quickly on the specific site. I am surprised that there is no reference to synthesis papers on end-to-end modeling . I did a quick search to see if such papers exist, and (without trying to be comprehensive) found Rose et al. 2010, “End-To-End Models for the Analysis of Marine Ecosystems: Challenges, Issues, and Next Steps,” DOI: 10.1577/C09-059.1. This, and perhaps other synthesis papers if available, need to be discussed.

I agree with reviewer 2 that the paper requires major revision. That revision needs to take into account my previous paragraph, as well as the comments from both reviewers. A thorough response to those comments (whether the authors choose to implement changes or refute the criticism) is needed.

Without prejudice of others, particular attention needs to be paid in the revisions to structure, breadth (for the Introduction), methodological detail, and methodological options that are questioned by the reviewers (e.g., the nine-year cycle for the simulations).

Reviewer 1 ·

Basic reporting

No comment

Experimental design

I do not believe that the methods are described with quite enough detail - see comments for the author.

Validity of the findings

I think the base model used would be better if driven by a repeating annual cycle of oceanographic data - see comments for the author. But the validity of the findings seem sound as is.

Additional comments

General remarks
My overall impression is that this is a thorough, well constructed analyses, with good presentation. I recommend it for publication with minor revisions.

I do not believe that modifications to the methods are necessary for publication, but I think that more explanation of some methods is needed to aid reader’s understanding and for reproducibility.
More discussion would be beneficial in places.
Also, ensure that all modelling choices (parametrisations/functional responses) are available to the readers – I think some things are lacking at present, eg predator-prey functional responses.

The authors use the Atlantis modelling framework to simulate an ecosystem east of New Zealand. I have some familiarity with ecosystem models, but have not used Atlantis. This made following the methods a little troublesome as much of the detail is left to references.

Fisheries management considerations is a focus of the introduction that is not really returned to during the discussion. More discussion of the potential of the model to inform management strategies would be good.

Atlantis is a spatial model, but spatial results were not reported. Asking for a breakdown of spatial results would involve a lot of extra work that is not necessary for this paper. But I think that the spatial aspect should be highlighted in the discussion of potential future work, particularly with regard to fisheries management.

I have only one potentially important methodological concern: the repeating cycle of ROMS years used for the base model. It seems to me that removing inter-annual oceanographic variability by using the average over all years would be preferable to repeating a cycle of 9 years.

See detailed comments below. Most of these comments are simply asking for clarification or more detailed discussion – some are simply questions not necessarily requiring changes to the paper.
Apologies for the review being rather long-winded, but I was surprised by the requested 10 day return time which I found too short to produce a concise review summarising the below comments.

Abstract
Well structured and informative.
As you mention ecosystem based approach to fisheries management I was surprised that model applications in the final sentence did not include using the model for forecasting biomasses under various fishing regimes. If this is a potential application(?) then mentioning it (or something similar) would nicely link back to fisheries management.

Introduction
Well structured interesting intro, gives sufficient context and background information on the study system, very brief synopsis of Atlantis modelling framework, clearly states objectives.

First paragraph says more ecosystem-based approaches are desired by fisheries management, and states the importance of Chatham Rise fisheries. The focus at the very start of the paper is fisheries management. I think that the potential of the model to guide fisheries management could feature more in the discussion section, as the focus seems to drift away from fisheries management.

Second Paragraph summarises the range of habitats and depth, and influence of subtropical front. The results are not presented in these terms, nor does the discussion section return to these factors. In the discussion more mention of potential for interesting spatial results should be included.

Line 79. You mention how ecological changes may influence the distribution of species, and that knowledge of this can be useful for management. The Atlantis model is spatially resolved into 23 compartments. Were there interesting spatial results? Eg were species distributions linked to warm/cold years, or fishing? I don’t know how much extra work would be involved in such analyses… if it’s substantial maybe just mention as potential future application, but if spatial results are readily available and easily analysed then more findings could be reported here…

Line 103. The word response seems dubious here, can you rephrase this line? Section 5 compares trends in model predicted catch to trends in observed catch and CPUE indices. Catch is the result of a certain fishing regime, but wouldn’t response to fishing be tested by examining modelled catch and abundance under various fishing regimes?

Model area
Lines 122-149. Lots of detail from existing studies about species richness-at-depth is presented here. There is more detail than necessary given that modelled spatial (depth) distributions were not reported/analysed in this study. The paragraph could be reduced to lines 122-126 and 139-141, then the focus would be very much on the polygon arrangement instead of extra detail (which could be retained in discussion section) .

Line 142. serve as a discontinuity is a bit cryptic – could be rephrased. Or, given the previous comment, this detail could be removed from the model area section.

Line 155. Model polygons are based on survey strata boundaries. This makes sense as it will make it relatively easy to assess model performance in terms of spatial distributions and species richness in different areas. Perhaps this should be mentioned…

Oceanography
Line 182. ‘The base model presented here repeated the available ROMS variables’. This is too little detail. Were the full ROMS time series repeated in a 9 year cycle? Were the ROMS data averaged over years then used in an annually repeating cycle? What exactly was done?

Figure 3 (right-hand plot). Seems an odd choice of partition for years given that the ROMS years (blue line) are 1996-2004. The yellow line is pre-ROMS years; whereas the black dashed line is some pre-ROMS years, the ROMS years, and post-ROMS years – which is not useful. Please change this so that the black dashed line shows temperature only in post-ROMS years. That way we can see how ROMS years compare with the past and the future temperatures.

Nutrients
Figure 4. This is not an important request, but it would be good if you could change the colour of the arrows because thin blue arrows on a blue/green background look fine on a computer screen but are not very clear on print-outs.

Line 192-193. What were the choices of initial values, and are these available?

Line 204. NIWA acronym should be written in full in first instance. I realise that three of the authors are based at this institution and that the acronym is spelled out on the title page, but as that is not the main text readers are unlikely to recall NIWA by line 204 (I didn’t).

Line 205. A small quibble here: can ‘ammonium is fairly small in terms of the nitrogen budget’ be rephrased to something like ‘ammonium is a small component of the nitrogen budget’?

Species groups – initial condition & parameters
Room for lots of uncertainty in specifying the initial conditions. The model is run on from these initials to approach a steady state before any analysed output is generated, so hopefully uncertainty in the initials is not too problematic… Did you test whether using different initial states influenced model behaviour? It would involve perturbing the initial conditions (possibly strong perturbations if uncertainties in initials are large) then running the model through the spin-up years to see if it approaches a steady state similar to the original. I suspect that the results may be sensitive to the initial states, particularly as there are so many, 53, functional groups. So some discussion on sensitivity to initial state values would be useful. I suppose this ties in with the discussion on knowledge gaps…

Lines 220-223. Fair enough, some sort of method needs deployed to get at initial biomass values. But were these initial values for the entire Chatham Rise, or were they calculated separately for each model polygon? Please explain how initial biomasses were specified for each polygon and depth layer. Also, are these initial values (which are also modelling results) available?

Lines 223-226. A bit more explanation would be useful as I’m unclear about how an absolute abundance estimate for a single year, 2003, was calculated from mean relative abundance over 23 years, 1992-2014. Also, if the authors decided what catchability quotient to use then please write ‘… derived by our expert opinion… ’, otherwise credit the expert with a reference.

Line 229. If it is the author’s expert opinion then please write ‘… our expert opinion’, otherwise credit the expert with a reference.

Table 6 & Appendix B. Here I’m confused by the von Bertalanffy growth parameters. It seems as though the values in the table are only used to initialise weights-at-age, because Appendix B shows that the model outputs different size-at-age from this table. I think this must mean that Atlantis is free to vary the von Bertalanffy parameters, but I also thought that parameter tuning was infeasible due to the complexity of Atlantis… so what’s happening here? How does Atlantis return size-at-age estimates?

Predation
Plaganyi et al. (2007) indicates that 6 predator-prey functional responses are available to use in Atlantis. Why is there is no mention of these functions in this study? The choice of functional response should be made clear.

Figure 5 is a good summary, but as you mention, it lacks detail. The supplementary material should contain a table of numeric values displaying each consumer’s diet as proportions (summing to 1) of individual prey species. Obviously this table would still lack spatio-temporal variation, but at least we could see each prey species.

It is confusing that the description of figure 5 contains the first mention of the ‘fished model’. If you run the model with and without fishing then this needs to be mentioned in the main text before reference to it appears in figure descriptions.

Calibration
Line 265 (and throughout). The syntax used for species group suffixes changes between dashes and brackets, eg invertebrate scavengers – commercial on line 265, and invertebrate herbivores (commercial) on line 277. Can you use the same style throughout? I think the brackets are better for readability.

Line 266. Although I don’t think it necessary for publication, this could be tested by removing the inter-annual variability from the oceanography data by averaging the data over years to create a repeating annual cycle to drive the model (see comment on line 182).

Line 268 (and throughout). There is repeated mention of Appendix 10 throughout the paper. But I don’t see an Appendix 10. Biomass trajectories (line 268) are displayed in Appendix A.

Lines 271-282. I agree that most of growth trajectory plots coincide quite well. But I don’t understand why the model grows species according to von Bertalanffy parameters that differ from Table 6 (see previous comment). Line 506 states that Atlantis is too complex to fit to data. I understand this to mean that Atlantis does not tune the input parameters to take different values, instead it simply runs the model using a fixed set of parameters. This needs more explanation.

Lines 271-282. The sizes of young animals are often overestimated. I think that this anomaly merits further investigation as it may be due to overconsumption of small individuals by some predator group(s). A more detailed analysis of the predator-prey relationships (age-based and spatial) could be useful here. This is probably best left for future work, but please elaborate on possible causes of the biased sizes at young ages in the discussion section.

Lines 294-301. Why was 2016 chosen? The biomass trophic level relationship should be fairly similar across years. Can you perform this calculation for each year and report the minimum, maximum and mean slopes.

Sensitivity analyses – oceanography
Lines 311-323. I like this section, it’s a nice way to test model sensitivity to variable oceanography. I’m not asking for changes to this section but it has made me wonder, again, about the driving data used in the base model (see previous comments). Surely it would be best for the base model to use an annually repeating driving data cycle calculated as the mean over all ROMS years (this could also be used instead of 2003 data for the burn-in period in these sensitivity simulations).

Line 324. Please mention the metric that these confidence intervals apply to. I assume it’s biomasses…

Lines 324-328. This is an interesting paragraph that could benefit from an extra sentence of explanation. Here the species groups with the highest CVs are those deemed most sensitive to oceanographic variability(?) - and I was not surprised by the species groups listed, the sensibleness of these results could be mentioned.

Lines 329-332. As with comment on line 324, can you rephrase ‘… that went above/below the established confidence intervals…’ to mention the actual metric used.

Lines 329-337. This is a nice way to illustrate effects of warm/cold years. A little more detail could easily be provided. Can you also mention whether it was the same species that fared better or worse during the warm & cold years. For example, during the 3 warmest years was there variation in the species that fared worse or was it the same species each year?

Connectivity and influence
Lines 348-366. The difference between Libralato et al. (2006) and your method is the extra term involving m in equation 4, and possibly a different definition of m (equation 2). I don’t understand the purpose of the m term in equation 4. If you have modified the method then please explain how and why.

Fishing
Lines 380-391. Please make it clearer how the catch data are used by the model. It seems that the catch is driving data used similarly to the oceanography metrics, but this is only alluded to in the description in Appendix D which mentions ‘forced catch history’.

Line 387. No request for a change here, just wondering if the marked decline in orange roughy was due to long life span/long generation times or fishing out the large/most fecund individuals?

Comparison with CPUE and stock assessment
Figure 13 shows that biomass trends returned by Atlantis are similar to some biomass trends estimated from stock assessments and CPUE. Can you also report comparisons of the magnitude of biomasses estimated from these 3 methods, along with the trends? The description of figure 13 states that the stock assessment and CPUE metrics are scaled, so we do not know how magnitudes compare…

Skill assessment
Lines 422-425. This little section may be confusing. Make it totally clear that n is year in equations 6-8, and that ‘point estimates’ are annual biomass estimates. Also change Appendix 10 to Appendix D in line 421.

Line 429. Mention that the MEF of ling is so low because of the stationary nature of the biomass estimates (displayed in Appendix D). MEF tests whether the model performs better than the mean of the data – well in the case of ling, only the data point with the relatively small CI could differ significantly from the mean.

Line 435. Equation 7 is not simple. Can values of RI>1 be interpreted in a meaningful way for each species group, or is this metric useful only for cross species comparisons? A little more discussion on RI metric may help understanding (is RI=1.5 acceptable, or is that a poor result?).

Lines 440-442. I agree with this sentence, and am not requesting changes to the analyses. But it made me wonder why the highly uncertain survey biomass estimates were used instead of stock assessment estimates. I realise that the survey contains more species, but here stronger results may be available through use of the stock assessments. Can you explain why the surveys were chosen over the stock assessments?

Bringing it together
Lines 448-450. Why is poor knowledge not concerning if paired with high responsiveness? Please provide a reference or explanation. Also, surely the ‘double’ of high keystoneness and low information (eg seabirds) are important considerations for future scenarios.

Lines 461-463. Why could this be? Please include brief discussion of potential causes. Mortality is size based so maybe something is eating most of the small individuals. The last part of this sentence, ‘… so this is probably not influential on the model over all’, is dubious and should be removed unless it can be backed up.

Lines 469-472. It seems that spiny dogfish are well modelled, but abundance is uncertain. There’s lots of them and they have high keystoneness, so it’s worth checking if they are linked to other anomalies like the small pelagic fish mentioned above. It would be good so see this kind of investigation into the causes of the mentioned anomalies. Perhaps that is best left for future work, but here a brief mention of that kind of analyses would be good.

Lines 477-483. Agreed. Data on these groups is often scant… which is frustrating as you have nicely demonstrated their importance in the model dynamics.

Discussion
Line 506. See previous comment. If model is too complex to fit to data, why does the model return growth trajectories that differ from the von Bertalanffy curves (Table 6 & Appendix B). Are the discrepancies entirely due to mortality at size/age? If so, then this should be mentioned (and it would seem that for some species the smallest animals from young age groups are being over-consumed within the model).

Lines 513-517. Agreed. But as ‘the influence of species groups on the rest’ (line 509) are key to developing understanding, digging deeper to see exactly what each modelled species is eating may expose where the model is not behaving realistically. Maybe mention this as potential future work.

Lines 525-527. Solution (a): does data currently exist that would enable this? If so, then mention why has it not been used here? If not, then only solution (b) is currently viable…

Conclusions
Line 547. Remove the word ‘very’. It doesn’t add anything here, and I’m unsure about close the biomass estimates were to each other – it was only the trends that were compared, not the actual values.

Reviewer 2 ·

Basic reporting

General comment:
In the manuscript "From data compilation to model validation: A comprehensive analysis of a full deep-sea ecosystem model of the Chatham Rise" the authors present and discuss the application of an end-to-end ecosystem model, the Atlantis model, in the Chatham Rise ecosystem.

Overall the manuscript and the research presented are relevant and well contextualized. The English language is overall good and clear for an international audience. However, the manuscript would benefit of some revisions before acceptance, in particular regarding its structure, in order to improve its clarity.

Specific comments:
1. The authors begin Introduction by describing the specific situation in New Zealand and the Chatham Rise study case. This could be improved if the authors provided a broader background/context of the problem addressed first (e.g. background on ecosystem management) and then described the specific case of the Chatham Rise ecosystem.

2. The authors do not follow a strict Introduction, Materials and Methods, Results, Discussion, Conclusions structure. Instead, they describe their methods and present simultaneously their results through sections 3 to 7. Section 2 refers to Model design.
This structure is acceptable but the paper would greatly benefit of a prior section (e.g. section 2 - Methodological Approach) where the authors provided an general overview of the methodological approach used, which would help the reader to follow throughout the manuscript. Although this is partly done in the the end of the Introduction (lines 95-107) it could be improved for clarity.

3. Section 2 - Model design could be improved by including a brief description and general scheme of the Atlantis model.

4. Figures and tables are overall clear and of good quality, but some improvements can be made. In particular, the legibility in black and white of some figures can be improved (e.g. Figure 9, Figure 15). Figure 1 could also be improved by including a scale, coordinates and North arrow. The acronyms used in Table 1 should also be in the table legend.

5. The authors provided their raw data and the model code, which I acknowledge. However, some additional description of the content of these files could be useful for future readers.

6. The Appendixes are referenced throughout the manuscript as Appendix 10. Please revise.

7. Abstract, line 36. I would suggest to add "which is also to be best of our knowledge the first end-to-end ecosystem model of any deep-sea ecosystem"

Experimental design

General comment:
The research presented in the manuscript is within the scope of the journal.

Overall the research question is well defined and the experimental design is adequate to the aims of the manuscript.

Specific comments:
1. Lines 172-173. The authors refer that used a 12 hours time step. Did the authors performed any sensitivity analysis of the time step? How does the time affect the model results?

2. Section 2.3. Further information or references are required regarding the ROMS application used in the present study.

3. Please clarify if the scheme presented in Figure 4 is a general overview of the Nitrogen cycle of if it also refers to the processes considered within Atlantis model.

Validity of the findings

General comment:
In this manuscript the authors did a great effort in establishing a complex ecosystem model and the findings are supported by the methodological approach followed. This model is intended to be used and improved in future applications regarding the fisheries management.
The main weakness of the model is probably its complexity, which difficult its validation. This limitation is acknowledge by the authors by referring that the model require further exploration

Specific comments
1. Please clarify the text in lines 334-337, 521-522, 528.

2. Please clarify which were the oceanographic conditions used in the model skill assessment simulations, since the the ROMS data do not cover the entire period.

3. The discussion of the results in section 6 - Skill assessment could be improved, in particular by detailing and improving the justification between the differences found.

4. Conclusions could be improved by including part of the discussion presented in lines 537-545.

Additional comments

I acknowledge the authors for their great effort in establishing such a complex model.
My recommendation regarding Major revisions of the manuscript in mainly related with its structure, since in my opinion the manuscript would benefit if the Methodological Approach was clearly described to help the reader to follow through the remaining sections.

---

## Round 0.2 · accepted · Accept

I concur with the reviewers that the revision addresses earlier comments, and is now deserving of publication. Congratulations.

# Reviewer 1 ·

Basic reporting

no comment

Experimental design

no comment

Validity of the findings

no comment

Additional comments

I am happy to recommend this manuscript for publication as the authors have met my concerns with the original submission in their revisions. The methods/results appear unchanged, but the additional text provides more detailed discussion and sufficiently explains modelling choices and aspects of the Atlantis model framework that I previously found confusing. I agree with the other reviewer that the addition of the Methodological Approach section improves the clarity of the paper – it certainly helped me as a reader.

Reviewer 2 ·

Basic reporting

The authors did a good effort in improving the manuscript taking into account the reviewers' comments and suggestions.
The addition of section 2 helps the reader to follow the manuscript.
Clarifications to the methods and approach used were provided.
Clarifications to the text were made when requested.
The discussion was improved.
Some figures and tables were also improved.

Experimental design

No comment.

Validity of the findings

No comment.

Additional comments

I acknowledge the authors for the review of the manuscript which, in my opinion, is suitable for publication.